# Regulation of Autophagy via Carbohydrate and Lipid Metabolism in Cancer

**DOI:** 10.3390/cancers15082195

**Published:** 2023-04-07

**Authors:** Javad Alizadeh, Mahboubeh Kavoosi, Navjit Singh, Shahrokh Lorzadeh, Amir Ravandi, Biniam Kidane, Naseer Ahmed, Fatima Mraiche, Michael R. Mowat, Saeid Ghavami

**Affiliations:** 1Department of Human Anatomy and Cell Science, College of Medicine, University of Manitoba, Winnipeg, MB R3E 0V9, Canadashahrokh.lorzadeh@umanitoba.ca (S.L.); 2Department of Physiology and Pathophysiology, Rady Faculty of Health Sciences, Institute of Cardiovascular Sciences, Albrechtsen Research Centre, St. Boniface Hospital, Winnipeg, MB R2H 2A6, Canada; amir.ravandi@umanitoba.ca; 3Section of Thoracic Surgery, Department of Surgery, Health Sciences Centre, Rady Faculty of Health Sciences, University of Manitoba, Winnipeg, MB R3T 6C5, Canada; bkidane@hsc.mb.ca; 4CancerCare Manitoba Research Institute, Winnipeg, MB R3E 0V9, Canada; nahmed2@cancercare.mb.ca (N.A.);; 5Department of Radiology, Section of Radiation Oncology, Rady Faculty of Health Sciences, University of Manitoba, Winnipeg, MB R3T 2N2, Canada; 6College of Pharmacy, QU Health, Qatar University, Doha 2713, Qatar; mraiche1@ualberta.ca; 7Department of Pharmacology, Faculty of Medicine and Dentistry, University of Alberta, Edmonton, AB T6G 2H7, Canada; 8Department of Biochemistry & Medical Genetics, University of Manitoba, Winnipeg, MB R3E 0J9, Canada; 9Research Institute of Oncology and Hematology, Winnipeg, MB R3E 0V9, Canada; 10Faculty of Medicine in Zabrze, Academia of Silesia, 41-800 Zabrze, Poland; 11Biology of Breathing Theme, Children Hospital Research Institute of Manitoba, University of Manitoba, Winnipeg, MB R3E 3P5, Canada

**Keywords:** mitophagy, non-small cell lung carcinoma, Bcl2 family protein, glycolysis, ceramide metabolism, Warburg effect

## Abstract

**Simple Summary:**

The metabolism of carbohydrates and lipids is essential to meet the diverse needs of cells. The autophagy cellular pathway is crucial for normal cellular function through its ability to target and degrade damaged organelles and misfolded proteins. Autophagy also provides cells with necessary energy as an end product. Cancer cells have a high rate of proliferation and, thus, require more energy compared to normal cells. Not surprisingly, cancer cells are highly dependent on these metabolic pathways. This review provides an in-depth review of recent research findings on how carbohydrate and lipid metabolism regulate autophagy and how this impacts cancer cells.

**Abstract:**

Metabolic changes are an important component of tumor cell progression. Tumor cells adapt to environmental stresses via changes to carbohydrate and lipid metabolism. Autophagy, a physiological process in mammalian cells that digests damaged organelles and misfolded proteins via lysosomal degradation, is closely associated with metabolism in mammalian cells, acting as a meter of cellular ATP levels. In this review, we discuss the changes in glycolytic and lipid biosynthetic pathways in mammalian cells and their impact on carcinogenesis via the autophagy pathway. In addition, we discuss the impact of these metabolic pathways on autophagy in lung cancer.

## 1. Introduction

Autophagy is a highly regulated process and generally classified into three major types: Macroautophagy, Microautophagy and Chaperone-Mediated Autophagy (CMA) (Figure 1A) [1,2,3]. Macroautophagy (thereafter autophagy) involves the formation of double membrane structures called autophagosomes, which engulf cargoes, such as misfolded proteins, cytoplasmic proteins, or damaged organelles within the cell.

The autophagosomes then fuse with lysosomes, which contain hydrolases, resulting in the degradation and recycling of the cargo’s contents [4,5,6]. Microautophagy is the direct fusion of cargo with lysosomes. During microautophagy, the lysosomal or endosomal membranes are invaginated followed by the fusion of the cargo with these organelles and finally their degradation and recycling [7]. CMA is a very selective type of autophagy. CMA is distinct from macroautophagy and microautophagy in that it does not require membranes for the engulfment of the cargo. CMA selectively targets proteins that contain a specific KFERQ pentapeptide residue (Lys-Phe-Glu-Arg-Gln). This residue is detected by the cytosolic heat shock cognate 70 KDa protein (hsc70). The target protein and hsc70 complex are transported into the lysosome by the lysosomal-associated membrane protein 2A (LAMP2A) receptor in the lysosomal membrane [8,9]. Autophagy can also be selective in detecting cargoes using specific mediators. Therefore, this has resulted in new terminologies, such as pexophagy (for peroxisomes), mitophagy (for mitochondria), and aggrephagy (for aggregates) [10,11] (Figure 1B). The mammalian target of rapamycin (mTOR) complex I (mTORC I) is the most well-known autophagy repressor [12,13]. Autophagy inducers such as starvation, inhibit mTORC I and facilitate the activation of ULK1 and autophagy-related 13 (ATG13) and, thus, inducing autophagy [14]. In addition, AMP kinase (AMPK) is activated upon low ATP/AMP ratios and is one of the main inhibitors of mTORC I that, consequently, induces autophagy [15,16]. It also activates autophagy through phosphorylating ULK1.

Autophagy protects cells from stress caused by DNA damage, reactive oxygen species (ROS), adenosine triphosphate (ATP) deficiency, and nutrient deprivation [17,18,19,20,21]. In normal cells, autophagy occurs at a basal level, whereas it significantly increases in cancer cells to support their high metabolic energy demands [22,23,24,25].

Autophagy is tightly regulated via different mechanisms. These include the processing of autophagy-related gene (ATG) proteins, the ULK1 kinase complex (ULK1-mAtg13–FIP200–Atg101), the class III phosphatidylinositol 3-kinase (PI3K) complex, VPS34 complex, and the ATG5–ATG12–ATG16 complex interaction with light chain 3 (LC3)-II [19,26,27] (Figure 1A). Autophagy is usually part of the response to chemotherapy agents in tumor cells and it can increase or decrease chemotherapy-induced apoptosis [28,29]. This response is dependent on the type of tumour cells and chemotherapy agents used [29,30,31,32,33]. On the other hand, autophagy-mediated cell death could also be involved in response to chemotherapy agents. Autophagy-dependent cell death, also known as autophagic cell death, is defined as a tightly regulated process, which does not require other cell death pathways. Autophagic-dependent cell death is mechanistically dependent on the autophagic machinery [34,35] and is characterized by the existence of a large number of cellular vacuoles (vacuolization), lysosomal degradation, no nuclear condensation and no caspase activation [12,13].

Autophagy plays a double-edged sword role in cancer-suppression and cancer-promotion [9]. In the cancer initiation stages, autophagy removes damaged organelles or misfolded proteins to suppress factors involved in cancer initiation (such as ROS and the unfolded protein response). Conversely, in late stages of cancer, autophagy may provide energy for cancer cell metabolism, facilitate tumor cell metastasis [36,37,38,39], or induce chemotherapy resistance [9,19,40].

### General Metabolism

Metabolism comprises all of the biochemical reactions that living cells use to generate the energy required for cell development, survival, and proliferation. [41,42,43,44]. Glycolysis, the Krebs cycle (glucose metabolism), and lipid metabolism are responsible for providing the required energy and building blocks for living cells. Glycolysis occurs in both aerobic and anaerobic conditions. The steps of the glycolysis pathway are summarized in Figure 2A. The Krebs cycle, also known as the citric acid cycle or the tricarboxylic acid (TCA) cycle, is summarized in Figure 2B. The interconnection of glycolysis and the TCA cycle is visualized in (Figure 2A,B). Fatty acids (FAs) are the most common form of stored energy, and triacylglycerols (TAGs) are the most common non-toxic form of FAs [45]. Generally, there are four sources from which to acquire FAs and TAGs: de novo lipogenesis (DNL), cytoplasmic TAG stores, FAs derived from TAGs of lipoprotein remnants taken up by the liver, and plasma non-esterified fatty acids (NEFA) released by adipose tissue [46]. The liver and adipose tissue are the two main sources in the body that produce FAs. FAs synthesized and exported from the liver provide an energy source and lipoprotein structural components for building cell membranes. FAs synthesized in adipose tissue through DNL contribute to long-term energy storage [47]. DNL is a metabolic pathway under tight regulation that involves a series of enzymatic reactions, occurring primarily in the liver and adipose tissue, to produce FAs from surplus carbohydrates to store energy. This pathway is summarized in Figure 2C [48,49,50].

In the presence of oxygen, pyruvate is transported into the mitochondria where it is converted into acetyl-CoA. Furthermore, many cells also utilize FA β-oxidation, which involves breaking down a long-chain acyl-CoA molecule into acetyl-CoA molecules (summarized in Figure 2D) [52,53,54,55,56]. The other lipid metabolic pathway is triglyceride biosynthesis. They are synthesized by the esterification of FAs into glycerol (summarized in Figure 2E) [47,57,58,59].

Compared to normal cells and apart from autophagy, cancer cells exhibit increased glucose uptake and lactate production even in the presence of oxygen [60]. This process of aerobic glycolysis was coined the “Warburg effect”, named after Otto Warburg, who first observed this [60]. Originally, Warburg hypothesized that this was due to a mitochondrial dysfunction in tumour cells but it has since been found that tumour cells and normal cells have similar rates of OXPHOS, suggesting that tumour cells have intact mitochondria [61]. However, if that is the case, why would tumour cells use this inefficient means of energy production?

The speed of cytosolic ATP production is about 100 times faster than in mitochondria. Therefore, aerobic glycolysis supplies more ATP per unit time than OXPHOS [62]. Cells with this higher rate of ATP production gain a selective advantage when accessing shared and limited resources such as the limited glucose in the tumour microenvironment [63]. When the demand for ATP is increased, aerobic glycolysis can be increased rapidly, while OXPHOS remains constant [62]. This supports malignant cell growth and proliferation. The Warburg effect also supports uncontrolled proliferation by feeding into biosynthesis. The increased glucose consumption serves as a carbon source for anabolic processes, such as nucleotide, lipid, and protein synthesis. The excess glucose also allows for a greater production of reducing equivalents (NADPH) through the pentose phosphate pathway (PPP), which is needed for reductive biosynthesis [61]. It has been proposed that this inefficient method of ATP production is a trade-off for maintaining the high flux through anabolic pathways.

The Warburg effect is of prominent importance in cancer cells for numerous reasons. It results in a higher rate of cytosolic ATP production compared with that in mitochondria. Therefore, aerobic glycolysis produces more ATP than oxidative phosphorylation (OXPHOS) per unit time [62]. Cells with this higher rate of ATP production gain a selective advantage when accessing shared and limited resources such as the limited glucose in the cancer microenvironment [64]. When the demand for ATP is increased, aerobic glycolysis is rapidly increased, while OXPHOS remains constant [62,64]. The Warburg effect also supports unregulated proliferation by feeding into biosynthesis. The increased glucose consumption serves as a carbon source for anabolic processes (e.g., lipid, nucleotide, and protein synthesis). The excess glucose also allows for a greater production of reducing equivalents of NADPH by the PPP, which is needed for reductive biosynthesis [64]. It has been proposed that this inefficient method of ATP production is a trade-off for maintaining the high flux through anabolic pathways in cancer cells.

Lactate may be the only oncometabolite involved in all the hallmarks of cancer progression, and it has been suggested that lactate production may be one of the main purposes of the Warburg effect [62,65]. Lactate excretion from cancer cells contributes to the production of an acidic cancer microenvironment, conferring a growth and metastatic advantage to cancer cells [66]. The low extracellular pH (pHe) promotes mutagenesis, chromosomal abnormalities, and genome instability [67], allowing cells to acquire mutations that favor malignant progression. The low pH stimulates cancer cells to secrete proteins that will degrade the extracellular matrix (ECM) and promote angiogenesis, creating an environment that favors intravasation, invasion, and metastasis [65,66]. The acidic microenvironment also represses lymphocytes, helping cancer cells with immune evasion [68]. This allows cancer cells to survive, metastasize, and colonize distant sites. Finally, lactate and the acidic microenvironment induce epithelial-to-mesenchymal transition (EMT) in cancer cells through the activation of autophagy and transforming growth factor *β* (TGF*β*) [38,39,69] expression; thus, increasing motility and invasiveness and, subsequently, metastasis [70,71]. Several investigations have shown that the suppression of glycolysis increases autophagy, which becomes a propelling force of OXPHOS for ATP production to support cancer cell survival [72,73,74]. Recent research has shown that glycolytic enzymes and the metabolic intermediates of glycolysis are involved in autophagy regulation [72,73,74].

The reprogramming of lipid metabolism in cancer directly contributes to malignant transformation and progression [60]. The expression levels of some of the crucial enzymes in lipid metabolism are altered in cancer. For example, fatty acid synthase (FASN), which is a multi-enzyme protein complex that catalyzes FA synthesis, is upregulated in cancer [64]. Sterol regulatory element-binding proteins (SREBPs) are a family of transcription factors regulating the expression of genes involved in the uptake and synthesis of FAs [71]. SREBPs are also upregulated in various cancers and promote cancer growth. It has been shown that the overexpression of lipid metabolism-related enzymes (e.g., FASN and SREBPs) is correlated with advanced stages of different type of cancers, poor patient survival, and cancer cell proliferation, invasion, and metastasis [75]. Moreover, the fatty acid oxidation (FAO) of lipids is increased in cancer cells to provide them with ATP to promote growth and survival under metabolic stress. In addition, FAO produces NADPH, which protects cancer cells from the oxidative stress of ROS [75]. Interestingly, different studies have recently shown the intricate link between autophagy and lipid homeostasis [76,77]. Autophagy plays a crucial role in the intracellular degradation of lipids [78,79,80]. This helps produce the precursors required for the biosynthesis of other lipids, repair damaged structures, and produce the energy necessary for the anabolic processes within the cell that require lipid building blocks [60]. Lipids play a crucial role in autophagic activity in multiple tissues. Therefore, they have important implications for diseases such as cancer [81,82]. The interconnection between lipid metabolism and autophagy is further highlighted by shared regulators, such as AMPK, mTORC1, TFEB, and PPARs [83,84]. Similar to glycolysis, lipid metabolism can exert its effects on autophagy during cancer through both lipid metabolites and the enzymes involved in lipid metabolism.

Despite the use of aerobic glycolysis for ATP production, cancer cells rely on functional mitochondria. The promotion of aerobic glycolysis is activated by pro-oncogenic mutations, such as K-RAS, c-MYC, and PIK3CA, not mitochondrial inactivation. Elimination of mitochondrial DNA (mtDNA) has been shown to reduce tumour growth rates and limit tumorigenesis [85,86]. Mutations of mtDNA has been found in several cancers, including renal adenocarcinoma, prostate cancer, bladder cancer, colorectal cancer, breast cancer, and head and neck tumours [87]. These mutations allow the altered cellular metabolism to promote malignant transformation. Mitochondria are vital for tumour progression and growth through their biosynthetic and bioenergetic effects [86]. Mitochondria are important for oxidation–reduction (redox) reactions, control the production of reactive oxygen species (ROS), sequester and release Ca^2+^, thereby controlling cytosolic Ca^2+^ levels, provide building blocks for biosynthetic pathways, and are involved in anterograde and retrograde signaling with the nucleus [87].

Given the importance of mitochondria in cancer, there must be a way to regulate their production. To prevent the accumulation of defective mitochondria, mitophagy (the selective autophagy of mitochondria) selectively degrades excessive or damaged mitochondria [85]. The knockdown of autophagy in mouse models of cancer leads to the build-up of defective mitochondria, impairing mitochondrial respiration and cell growth, and increasing cell death [87]. Similarly, autophagy has been found to be crucial for mitochondrial metabolism and tumour growth in KRAS-driven lung cancer and BRAF-driven cancers [87]. Having mitochondrial quality control is important for malignant growth by preserving the supply of functioning mitochondria [85]. Defective mitochondria provide a direct signal to mitophagy machinery for degradation, whereas mitochondrial synthesis is triggered by reduced cellular energy output [86].

In the current review article, we will discuss and explain how the different steps of glycolysis and lipid metabolism are connected to autophagy, and their impact on tumorigenesis and metastasis. In the last section, we will specifically focus our discussion on the impact of autophagy and these metabolic pathways in lung cancer.

## 2. Glycolysis and Autophagy Regulation in Cancer

In this section, we will discuss the key enzymes involved in regulating glycolysis and their effect on autophagy.

### 2.1. Hexokinase 2 (HK2)

HK2 catalyzes the initial step of glycolysis, which is the conversion of glucose to glucose 6-phosphate [88,89]. In mammals, there are four hexokinase isoforms: HK1, HK2, HK3, and HK4 [90]. Among the subtypes, HK2 is considered the most important one, which is overexpressed in various tumors including breast, lung, gastric and laryngeal squamous cancer, hepatocellular carcinoma (HCC), and renal cell carcinoma [91,92,93,94,95,96,97]. In addition to its critical role in glycolysis and autophagy, HK2 also facilitates apoptosis resistance and the induction of chemo-resistance in pancreatic, epithelial ovarian and breast cancers by increasing its expression or dimerization and interaction with voltage-dependent anion channels [98,99,100,101,102]. HK2 is overexpressed in ErbB2-driven breast cancer for tumor initiation and maintenance [103]. Further studies have shown that targeting HK2 via miR-143 inhibited cell proliferation and the Warburg effect in colon [104], breast [105], prostate [106], and renal cell cancers [96]. Overall, overexpression of HK2 results in an enhanced autophagic response. In contrast, suppression of autophagy is seen with an inhibition of HK2 (Figure 3A) [107].

### 2.2. Phosphofructo-Kinase/Fructose Biphosphatases (PFKFBs)

PFKFB controls the intracellular steady-state concentration of fructose 2,6-bisphosphate (F2,6P2) [108]. There are four isozymes of PFKFB (PFKFB1–4), which display different properties, including tissue specific expression and the response to kinases and phosphatases [109,110]. PFKFB3 and PFKFB4 are found overexpressed in breast [107,111,112,113,114,115,116,117], bladder [118,119], pancreatic [120,121], and gastric cancers [120,121]. Mechanistically, low levels of ATP leads to the phosphorylation of the C-terminal domain at Ser461 of PFKFB3 by the cellular energy sensor AMPK [122]. In addition, this condition induces MK2 (MAPK (mitogen-activated protein kinase)-activated protein kinase 2) phosphorylation at Thr334 via p38α MAPK, which, in turn, activates PFKFB3 (phosphorylation at Ser461). Moreover, MK2 also activates SRE (serum-response element) in the promoter region of the PFKFB3 gene via the phosphorylation of SRF (serum response factor) at Ser103, resulting in the increased transcription and activity of PFKFB3 to amplified glycolytic and autophagic rates (Figure 3B) [122]. Klarer et al. demonstrated that the inhibition of PFKFB3 using either siRNA transfection or derivatives of the PFKFB3 inhibitor, 3-(3-pyridinyl)-1-(4-pyridinyl)-2-propen-1-one (3PO), initiates survival autophagy in HCT-116 colon adenocarcinoma cells [123]. Some studies showed that PFKFB4 depletion promoted oxidative stress, ultimately triggering survival autophagy as a stress-adaptive survival mechanism in lung cancer [124,125].

### 2.3. Glyceraldehyde-3-Phosphate Dehydrogenase (GAPDH)

GAPDH catalyzes the conversion of glyceraldehyde-3-phosphate to 1.3-bisphosphoglycerate in glycolysis, which is a reversible reaction [126,127]. As well as playing the vital role of maintaining aerobic glycolysis in the cytosol, GAPDH also participates in several non-glycolytic functions including autophagy [128]. In the cytosol, under glucose shortage, GAPDH is phosphorylated at Ser122 by AMPK. This leads to GAPDH redistributing into the nucleus, where it interacts directly with sirtuin1 (SIRT1) to stimulate SIRT1 activation and autophagy, increasing SIRT1 deacetylase activity and disassociating its inhibitor DBC1 (deleted in breast cancer-1) (Figure 3C) [129]. In general, various studies have shown that the GAPDH overexpression is associated with breast, ovarian, lung, and prostate cancers’ aggressiveness and metastases [126,130,131,132,133,134]. Therefore, GAPDH depletion is recommended as a novel strategy to control tumor cells. The GAPDH inhibitor (3-bromo-isoxazoline) induces autophagy and inhibits the proliferation of pancreatic ductal-adenocarcinoma cells (PDAC) and pancreatic cancer stem cells (CSCs) [135]. Furthermore, the ROS/Akt/mTOR axis is activated by the inhibition of mitochondrial uncoupling protein 2 (UCP2) in pancreatic adenocarcinoma cells, which, in turn, leads to tumor cell growth arrest, induction of caspase-mediated apoptosis, and the translocation of GAPDH into the nucleus to promote BECLIN1-mediated autophagy [136].

### 2.4. Phosphoglycerate Kinase (PGK)

PGK coordinates ATP production in glycolysis by catalyzing the conversion of 1,3-bisphosphoglycerate to 3-phosphoglycerate. PGK has two isoforms, PGK1 and PGK2 [137,138]. PGK1 has a significant role in sustaining cellular homeostasis and autophagy, making it an attractive molecular target for cancer therapy. The condition of glutamine deprivation and hypoxia inhibit mTOR-mediated arrest-defective protein 1 (ARD1) Ser228 phosphorylation, leading to ARD1 binding to PGK1 and Lys388 acetylation with subsequent PGK1-mediated BECLIN1 Ser30 phosphorylation. This activates ATG14L-associated class III phosphatidylinositol 3-kinase VPS34 to initiate autophagosome formation in glioblastoma (GBM) patients. ARD1-dependent PGK1 acetylation and PGK1-mediated BECLIN1 Ser30 phosphorylation are involved in brain tumorigenesis [139] (Figure 3D). In another investigation, it has been shown that PGK1 induced the phosphorylation of PRAS40 (the proline-rich AKT substrate of 40 kDa, encoded by the gene AKT1S1) at Thr246 in Ewing’s sarcoma cell line A673, and liver cancer cell lines HepG2 and SNU449. PRAS40 is involved in PGK1-induced cancer cell growth. It was found that PGK1 inhibits autophagy-mediated cell death, which is associated with PRAS40. In normoxic conditions, PGK1 blocks autophagic cell death via PRAS40 phosphorylation, stimulating cellular proliferation. In hypoxia, PGK1 interacts with BECLIN1 and induces its phosphorylation with subsequent autophagy induction. Therefore, PGK1 is, potentially, a novel autophagy modulator for tumorigenesis that acts between repressing autophagy-mediated cell death via PRAS40 and inducing autophagy through BECLIN1 based on oxygen levels in the environment [140].

### 2.5. Pyruvate Kinase (PK)

PK catalyzes the conversion of phosphoenolpyruvate (PEP) to pyruvate, which is the final step of glycolysis. PK has four isoforms: PKL (liver), PKR (erythrocytes), PKM1 (muscle and brain), and PKM2 (brain and liver) [141,142,143]. PKM2 can be a tetramer or dimer and the tetrameric form has a higher affinity for PEP, while the dimer has a lower affinity. Under physiological conditions, the tetramer form dominates, whereas cancer cells primarily prefer the dimeric form, which is necessary for the Warburg effect (aerobic glycolysis). In other words, the lower activity of the dimeric form of PKM2 induces an increase in glycolysis, and can promote tumor survival and growth [141,142,144,145,146,147,148,149]. Therefore, identifying new compounds that act as PKM2 activators could be a potential therapeutic strategy for cancer treatment.

Many investigations provide evidence that PKM2 is involved in autophagy. Overexpression of PKM2 leads to the activation of class I PI3K that could lead to activation of the Akt-mTOR signaling pathway, which, in turn, inhibits autophagy. PKM2 phosphorylates PRAS40 and releases this protein from mTORC1 with its subsequent activation in pancreatic cancer (Figure 3E) [150]. The knockdown of PKM2 in cancer cells activates AMPK signaling to maintain energy homeostasis via autophagy activation in prostate cancer cells [144]. The downregulation of PKM2 acts as a tumor suppressor in acute myeloid leukemia (AML) by increasing BECLIN1-mediated autophagy [150]. PKM2 is also overexpressed in HCC cells and is induced through the JAK/STAT3 pathway, which might be involved in the response of HCC tumor cells to chemotherapy-induced apoptosis [151].

### 2.6. Lactate Dehydrogenase (LDH)

LDH catalyzes the reduction of pyruvate to lactate with the associated interconversion of NADH and NAD+. It has two isomers, including LDHA and LDHB, that support the conversion of pyruvate to lactate in cells and back, respectively [152,153,154]. LDHA is enhanced in many types of cancers (lung, liver, testicular, and breast) to establish metabolic reprogramming [155,156,157,158,159]. In tamoxifen-resistant breast cancer cells, ATP production decreased in parallel with increased glycolytic pathway activity [160]. In this scenario, LDHA is associated with tamoxifen resistance via BECLIN1-induced pro-survival autophagy [160].

LDHB is a crucial contributor to lysosomal activity and basal autophagy in cancer. This enzyme converts lactate and NAD+ to pyruvate, NADH and H+ to create an acidic environment. This is essential for vesicle maturation and protease activation during autophagy; however, the regulation of this has not yet been defined [161]. Liang Shi et al. demonstrated that the protein sirtuin5 (SIRT5) caused hyperactivity of LDHB by the deacetylation at Lys329 that triggered autophagy and accelerated the growth of colorectal cancer cells (CRC) [150].

### 2.7. Lactate

One of the most important consequences of the Warburg effect in cancer cells is the increase in lactate production, which, in turn, leads to the acidification of the tumor microenvironment, creating an extracellular pH ranging from 6.0 to 6.5 (Figure 4) [162,163].

Emerging studies have shown that lactate-induced acidosis leads to processes such as metastasis [164,165], angiogenesis [166,167,168], invasion [165,169], and immunosuppression [162,170]. Lactate itself cannot freely diffuse across the plasma membrane and monocarboxylate transporters 1–4 (MCT1-4) control the exchange of lactate across membranes [171,172]. Among the MCTs, MCT1 and MCT4 are extensively expressed in cancer cell lines, playing an important role in metabolic reprogramming and tumor aggression [173,174,175,176,177]. Regulating the expression of MCT1 via Osimertinib (OSI) activates liver kinase B1 (LKB1)/AMPK signaling, leading to autophagy induction in CRC cells [178]. On the other hand, blocking lactic acid export by inhibiting MCT4 provides an efficient anticancer strategy. MCT4 depletion by different doses of 7-aminocarboxycoumarin derivatives (7acc1) enhanced the cytotoxicity of NK cells in breast carcinoma by interfering with lactate flux, reversing the acidification of the tumor microenvironment, and inducing autophagy [179].

### 2.8. Hypoxia-Inducible Factor-1 Alpha (HIF-1α) and Glycolysis

Adaptation to hypoxic conditions is achieved primarily through excessive accumulation of HIF-1α in tumor cells. HIF-1α increases the expression of the glycolysis pathway and compensates for the ATP demand of eukaryotic cells under hypoxic conditions [180]. Beyond its role in glycolysis, HIF-1α participates in autophagy by increasing the transcription of BNIP3 and BNIP3L genes, which leads to the dissociation of the Bcl-XL and Bcl-2 complex and the activation of BECLIN1-induced autophagy [181,182] (Figure 4). Notably, glucose deprivation could induce HIF1α-independent autophagy through the induction of AMPK and the inhibition of mTOR (Figure 3F) [183,184]. The mTOR signaling pathway is inactivated under hypoxic conditions, leading to activation of ULK1, which is required for the initiation of autophagy (Figure 3F). Therefore, hypoxia-induced autophagy may be considered as a target to reverse the radio resistance in cancer cells. Yueming Shen et al. showed that autophagy was inhibited by the knockdown of lincRNA-p21. This occurred due to the downregulation of HIF-1α protein levels and activation of the Akt/mTOR/P70S6K signaling pathways. This is a suggested mechanism to enhance the radiosensitivity of hypoxic tumor cells in human HCC and glioma cells [185]. A recent survey indicated that the expression of HIF-1α is commonly increased in a variety of human tumors, including lung, breast, and osteosarcoma [186].

We have summarized the impact of the glycolysis-related enzymes involved in the regulation of autophagy in different cancers (excluding lung cancer, which is discussed separately below) in Table 1.

## 3. Lipid Metabolism and Regulation of Autophagy in Cancer

Historically, lipids have been challenging to study. However, the advancements in technology and methodologies have made progress in the research on the role of lipids in autophagy and cancer progression possible. Despite recent discoveries about this relationship, our understanding of how lipid metabolism affects autophagy alterations during cancer is still rather limited. In this section, we aim to provide the most current findings and understanding of how changes in lipid metabolism influence autophagy during cancer. Lipids play a crucial role in autophagic activity in multiple tissues; therefore, they have important implications for diseases such as cancer [78,190]. The connection between lipid metabolism and autophagy is further strengthened by the common regulators that they both share, including AMPK, mTORC1, Transcription Factor EB (TFEB), and PPARs [191,192,193,194]. Autophagy during cancer progression is modulated thorough lipid metabolism.

### 3.1. Enzymes in Lipid Metabolism Can Regulate Autophagy in Different Cancers

#### 3.1.1. Fatty Acid Synthase (FASN)

FASN is an important cytoplasmic enzyme involved in fatty acid biosynthesis. The main function of FASN is to synthesize palmitate from acetyl-CoA and malonyl-CoA [195]. FASN expression increases the activation of mTOR, resulting in autophagy inhibition in leukemic blast cells obtained from AML patients [196]. Moreover, FASN can downregulate the expression of TFEB via mTOR. TFEB is phosphorylated by mTOR, leading to TFEB becoming localized in the cytoplasm and, thereby, inhibiting its transcriptional activity [196]. This can negatively affect autophagy induction since TFEB regulates the expression of the genes involved in autophagy including lysosomal biogenesis, lysosomal membrane, lysosomal acidification, and other autophagy-related genes [197]. This has been confirmed by the finding that the inhibition of FASN promotes the nuclear translocation of TFEB where it can positively regulate the expression of lysosomal biogenesis genes such as LAMP1 [196]. Additionally, FASN has been found to self-modulate its expression via crosstalk with PI3K/Akt in clinical specimens of human osteosarcomas. It was found that the overexpression of FASN has a positive relationship with the activation of PI3K-Akt in these clinical samples. In contrast, the inhibition of FASN decreased Akt phosphorylation while Akt inhibition similarly inhibited the mRNA and protein expression of FASN in vitro [198]. In another study, it was found that FASN overexpression enhanced the cellular respiration (e.g., FAO) that favors cancer cells during CRC. In addition, FASN overexpression was correlated with less MAPK activation and the accumulation of p62, indicating autophagy inhibition, which can favor the survival of CRC HCT116 and SW480 cells [199]. The overall effect of FAs in the regulation of autophagy is summarized in Figure 5.

#### 3.1.2. 3-Hydroxy-3-MethylGlutaryl-CoA Reductase (HMGCR) Inhibitors

HMGCR is the rate-limiting enzyme in the mevalonate pathway that synthesizes different lipids, such as cholesterol, geranylgeranyl pyrophosphate (GGPP), and farnesyl pyrophosphate (FPP) [28,29,200,201]. Statins are a class of medications that inhibit HMGCR and, thus, decrease the de novo cholesterol levels [200,202,203]. Therefore, statins have been commonly used as cholesterol-lowering agents. Statins have been shown to have pleiotropic effects that are independent of their cholesterol-lowering properties [201]. However, inhibition of HMGCR by statins can affect the production of other downstream lipids that are produced in the mevalonate cascade. Changes in these lipids, in turn, can modulate autophagy and, therefore, affect cancer cell survival. It has been shown in human leukemia cells that blocking cholesterol biosynthesis using statins leads to autophagy induction via inhibition of the Akt-mTOR pathway [204]. Interestingly, the addition of cholesterol (but not GGPP or FPP) to cells attenuates the induced autophagy, indicating the cholesterol-dependent inhibition of autophagy in these cancer cells [205].

In HepG2 cells, HMGCR inhibition by statins can induce autophagy through increasing LC3-II expression. Autophagy induction was confirmed after observing that statins further increased LC3-II expression when the cells were also treated with Bafilomycin A1, an autophagy inhibitor [206]. Similar results were observed in atorvastatin-treated PC3 prostate cancer cells, where a 10-fold increase in LC3-II expression was found. This group observed that the addition of GGPP, but not FPP, could reverse the atorvastatin-induced autophagy. This indicated that autophagy induction by atorvastatin was due to the inhibition of GGPP biosynthesis [207]. In human rhabdomyosarcoma cells, the depletion of GGPP with hydrophobic statins induced autophagy [208,209]. It was also shown that simvastatin inhibits the flux of autophagy in alveolar rhabdomyosarcoma cells through changing the acidity of lysosomes, which is different from C2C12 cells [210]. Statins (fluvastatin, lovastatin, simvastatin, and atorvastatin) can trigger the induction of autophagy in PC3 cells via the inhibition of geranylgeranylation. The findings of this study suggested that autophagy induction by statins can potentially have a protective role against prostate cancer progression [207,211]. These findings highlight that lipid metabolites produced by the mevalonate pathway play a crucial role in autophagy regulation upon HMGCR inhibition in different types of cancer. Additionally, it has been shown in previous studies that the combination therapy of statins with other anticancer agents can improve the treatment response in different cancer cell lines (small-cell lung, ovarian and breast cancer, GBM, and acute myeloid leukemia) [29,212,213,214,215]. In agreement with these studies, we have previously shown that simvastatin prevents the fusion of autophagosomes and lysosomes in GBM cell lines; thereby, inhibiting autophagy flux. This autophagy flux inhibition potentiates the temozolomide-induced apoptosis in GBM cancer cells [29]. The impact of cholesterol and the cholesterol biosynthesis pathway in the regulation of autophagy is summarized in Figure 6.

#### 3.1.3. Sphingosine-1-Kinase (SPHK1)

SPHK1 is the enzyme that synthesizes sphingosine-1-phosphate (S1P) from sphingosine. It also plays a crucial role in maintaining the sphingolipid balance within the cell [216]. It has been shown that SPHK1 participates in the regulation of autophagy and in different aspects of malignancy including therapy resistance, cancer development, growth, and metastasis [217,218]. SPHK1 is highly expressed in different types of cancers, such as human astrocytoma, CRC, and prostate cancer. It has also been associated with angiogenesis and resistance to chemotherapy and radiotherapy [219,220,221]. Thus, SPHK1 has been proposed to serve as an oncogene in carcinogenesis [222]. Prior studies have found that SPHK1 plays a role in the biogenesis of autophagosomes by modulating the expression levels of LC3 and, therefore, autophagy in neuroblastoma cells. However, the exact mechanisms by which SPHK1 regulates autophagy in these cancer cells has yet to be elucidated [223]. Another study found that in breast cancer MCF-7 cells, SPHK1 induces autophagy by increasing the formation of LC3-positive autophagosomes, thereby protecting them during nutrient deprivation [224]. SPHK1 also induces autophagy in CRC HT-29 cells via activation of SPHK1/ERK/pERK, which leads to the inhibition of mTOR and, consequently, ULK1 activation. SPHK1 was also shown to increase the expression of ATG5, ULK1, and LC3 in CRC HT-29 cells. Furthermore, autophagy has been linked to the survival of CRC cells during chemotherapy stress and is a marker of an invasive phenotype. Therefore, autophagy inhibitors, such as Bafilomycin A1 and Chloroquine, are suggested as promising interventions for CRC treatment [225]. Moreover, SPHK1 has been shown to induce EMT and, consequently, contribute to invasion and metastasis in HCC HepG2 cells [226]. This study revealed that SPHK1 induced EMT through enhancing lysosomal degradation of E-cadherin, a protein critical in epithelial cell–cell adhesion. Furthermore, this group found that S1P could stimulate the binding of TRAF2 (TNF Receptor Associated Factor 2) to BECLIN1. TRAF2 binding catalyzes the ubiquitination of BECLIN1 and, therefore, the induction of autophagy. Overall, the authors concluded that the SPHK1-E-cadherin-TRAF2-BECLIN1 pathway is responsible for EMT induction and subsequent metastasis in HCC cells. Thus, SPHK1 inhibition is proposed to decrease autophagy activation and is a possible strategy for HCC treatment. SPHK1 also modulates autophagy in human peritoneal mesothelial cells (HPMCs) and plays a role in the progression of gastric cancer peritoneal dissemination (GCPD) [227]. SPHK1 expression was directly related to LC3 expression in 120 human samples from gastric cancer peritoneal tissue. The SPHK1-induced autophagy in HPMCs increased the invasive capability of gastric cancer cells, whereas SPHK1 knockdown inhibited autophagy. This was associated with a poor prognosis and recurrence in these patients. SPHK1 is also involved in the production of ceramide, a prominent lipid metabolite that will be discussed in the following section. Additionally, SPHK1 has an inhibitory effect on mTOR and, therefore, allows for the activation of autophagy independent of Akt [81]. The connection of sphingomyelin, ceramide, and SPHK1 to autophagy is summarized in Figure 7.

#### 3.1.4. Fatty Acid Translocase/Cluster of Differentiation 36 (FAT/CD36)

FAT/CD36 is an enzyme localized within the cell membrane that is responsible for long-chain fatty acid uptake into the cell [228]. A study found that FAT overexpression inhibits autophagy in HCC HepG2 and Huh7 cells [229]. Consistent with these findings, they found that FAT knockdown induces autophagy in these cells as shown by the increased formation of autophagosomes, LC3-II expression, and p62 degradation. FAT modulates autophagy through the phosphorylation of BECLIN1 and ULK1, acting as a negative regulator of autophagy in HCC cells [229]. In this same study, in vivo studies in mice with FAT knockout also showed elevated autophagy. This was further confirmed by the observation that autophagy was decreased after reconstituting FAT expression in FAT knockdown mice. Additionally, FAT knockdown caused an increase in lipophagy levels together with enhanced FA β-oxidation. These findings highlight that modulation of FAT expression could be a promising way to induce lipophagy in HCC cells, thereby mitigating the excessive lipid accumulation (Figure 7).

We have summarized the effect of enzymes involved in lipid metabolism on the different steps of autophagy in Table 2.

### 3.2. Lipid Metabolites Can Regulate Autophagy in Different Cancers

#### 3.2.1. Phosphatidic Acid (PA)

PA is produced through the breakdown of phosphatidylcholine into PA and choline by Phospholipase D [230]. Although different targets for PA have been reported, the most significant target of PA in cancer cells is likely to be mTOR [231]. PA regulates autophagy by mediating the induction of autophagy by inducing autophagosome curvature and inhibiting mTOR [232]. Furthermore, PA can be produced under nutrient and oxygen deprivation, which is common in human tumor sites [231]. Other metabolites of PA can also regulate autophagy. For example, Phosphatidic Acid Phosphatases can convert PA into diacylglycerol, which has been shown to regulate autophagy [233]. Diacylglycerol induces autophagy through activation of PKC. It also helps dissociate BCL2 from BECLIN1 through activation of JNK and, thus, initiates autophagy induction [233]. Lysophosphatidic acid (LPA) activates mTORC1 and decreases autophagy via the phosphorylation of S6K and ULK in prostate cancer cells (PC3, LNCaP, and Du145 cells) [234]. It has been recently shown that the HS1BP3 protein negatively regulates autophagosome formation in moderately differentiated osteosarcoma cells (moderately differentiated sarcoma) through phospholipase D (PLD) and PA production and the increased localization of PLD1 to ATG16L1-positive membranes [235].

#### 3.2.2. Sphingolipids

Another class of lipids are sphingolipids, which play a prominent role in autophagy regulation as well as cell proliferation and differentiation [236]. Sphingolipids are a very important group of lipids that can regulate a wide range of crucial functions in cancer cells, such as cell growth and proliferation [237]. In particular, two sphingolipids have appeared to significantly regulate autophagy, ceramide and S1P. Ceramide can inhibit Akt, which consequently inhibits mTOR, inducing autophagy. It has also been found that both ceramide and S1P can upregulate the transcription of BECLIN1 and cause autophagy induction [238]. The product of ceramide hydrolysis is sphingosine, which produces S1P via phosphorylation by SPHK1 [224]. Recent studies have shown that SPHK1 is activated under starvation conditions, which consequently forms S1P [224]. Ceramide synthase 2 (CerS2) is one of six mammalian isoforms of ceramide synthase that can synthesize ceramide. It was shown that overexpression of CerS2 causes the accumulation of ceramides, resulting in growth arrest and autophagy induction in SMS-KCNR neuroblastoma and MCF-7 breast cancer cells [237]. Additionally, it has been shown that the treatment of HT-29 CRC cells with ceramide induces autophagy in these cells by elevating the pool of ceramides [238,239,240]. It was shown in MCF-7 breast cancer cells that ceramides increased the expression of BECLIN1 and, therefore, induced autophagy [239]. In addition, it has been shown that short-chain ceramides (C(2)-ceramide and C(6)-ceramide) and tamoxifen-induced stimulation of de novo ceramide synthesis caused the dissociation of BCL2 from BECLIN1 and, therefore, the induction of autophagy in HeLa cells [238].

Sphingosine is generated from the hydrolysis of ceramide by ceramidase followed by its conversion to Sphingosine-1-Phosphate (S1P) by sphingosine kinases. S1P binds to its cell receptors and can modulate various cellular functions including proliferation, migration, and autophagy. It has been shown that elevated levels of intracellular S1P through overexpression of SPHK1 induces autophagy via the inhibition of mTOR in human MCF-7 breast cancer cells [224,241]. Moreover, S1P can act as a survival mechanism in prostate cancer. It was shown that treatment with S1P in PC3 prostate cancer cells can induce autophagy through the inhibition of mTOR [242]. It was concluded that S1P could mediate nutrient deprivation-induced autophagy as a survival mechanism during the early phase of prostate cancer development.

#### 3.2.3. Peroxisome Proliferator-Activating Receptors (PPARs)

PPARs comprise a family of nuclear proteins including PPARα, PPARδ, and PPARγ that serve as transcription factors, regulating the expression of different genes [243]. In order to be functional as a transcription factor, PPARs need to bind to lipid ligands via lipid-binding proteins [244]. These lipid ligands have been reported to be oleic acid, linoleic acids, linolenic acids, prostaglandins, eicosanoids, and oxidized lipids [245,246]. PPARs have been shown to regulate autophagy in different cancers via various pathways such as the regulation of autophagy-related genes [81]. Furthermore, TFEB is also activated following the activation of PPARs, which upregulates the expression of different autophagy-related genes, such as Rab7 and LAMP2 [247]. This finding has been further confirmed by the observation that mouse cells with PPARδ knockout have a significant reduction in autophagy markers [248]. The levels of these lipid ligands change during cancer, which can directly influence PPAR and TFEB activation and, thereby, autophagy during cancer. For this reason, PPARs have been proposed as promising treatment candidates for cancer. It has been found that the treatment of Caco-2, a CRC cell line, with rosiglitazone (a PPARδ agonist) induces autophagy [249]. In addition, PPARγ can activate PTEN, which is an inhibitor of PI3K. Upon PI3K inhibition, PIP3 production is reduced, resulting in mTOR inhibition and the induction of autophagy in CRC cells [249]. PPARγ is active during cancer, and it has been suggested that PPARγ activates autophagy in a self-sufficient manner. For example, PPARγ has been shown to regulate autophagy in MDA-MB-231 breast cancer cells. It was shown that troglitazone (a PPARγ agonist) induces autophagy in these cancer cells as shown by an increase in lysosomal acidification [250]. These findings have instigated new proposals among cancer researchers that PPAR-induced autophagy benefits cancer cells by protecting them from metabolic stress and cancer therapy [250].

#### 3.2.4. Free Fatty Acids (FFAs)

FFAs are byproducts of lipophagy, and they can also regulate autophagy. The excessive accumulation of FFAs can result in cell death (known as lipotoxicity) [251]. Thus, the levels of FFAs are tightly regulated by lipophagy [252]. Palmitic acid is a common dietary FA [253]. A study on HCC cells showed that palmitic acid was able to induce autophagy flux independent of mTOR [254]. Human HCC HepG2 cells treated with palmitic acid showed activation of PKC and increased levels of diacylglycerol. As mentioned above, diacylglycerol can induce autophagy via the activation of PKC and the dissociation of BCL2 from BECLIN1. Furthermore, PKC inhibition significantly reduced autophagy flux in these cells, confirming that palmitic acid induced autophagy flux via PKC [254].

#### 3.2.5. Omega-3 Polyunsaturated Fatty Acids (n-3 PUFAs)

n-3 PUFAs are involved in the modulation of autophagy [255]. Docosahexaenoic acid (DHA) is one of the most studied n-3 PUFAs [256]. DHA can be synthesized in the body in low amounts through the conversion of α-linolenic acid (ALA), which is a shorter chain omega-3 fatty acid [257] by elongase [258,259]. DHA has antitumor effects on various types of cancer through the modulation of autophagy [260]. DHA has been studied in various cancers [261,262,263] and it exists in many phospholipids that form the cell membrane in different tissues [257]. In p53-wild type prostate cancer cell lines, it has been reported that DHA induces autophagy through AMPK activation resulting in the inhibition of mTOR [264]. In contrast, it induces autophagy through mitochondrial ROS production in p53-mutant prostate cancer cells (PC3 and DU145). The generated ROS lead to the inhibition of Akt and mTOR and, thus, autophagy induction [265]. Interestingly, DHA alone or in combination with other chemotherapeutic agents, can exert anticancer effects by boosting the immune response to cancer cells. It was found that DHA-induced autophagy increases the processing of cancer antigens. The subsequent cross-presentation of these cancer antigens to Antigen Presenting Cells (APCs) improved the immune response by ensuring the production of antitumor T cells [266]. DHA can also induce autophagy in CRC cell lines, Caco-2 and SW620 (by regulating the expression of genes involved in the autophagy process, such as p62, LC3 and ATG14) [267], and the HCC cell line, HepG2 (shown by the accumulation of LC3-II and the formation of LC3puncta) [268]. DHA exerts its anticancer effects in other cancer types, such as GBM and cervical cancer [264], by the induction of autophagy through mTOR inactivation.

DHA can be converted to docosahexaenoyl ethanolamine (DHEA) [269]. It has been shown that DHEA can activate autophagy in MCF-7 breast cancer cells [270]. It can increase the expression of PPARγ, which can, in turn, upregulate PTEN. Finally, PTEN inhibits Akt-mTOR, which subsequently induces autophagy. Moreover, in the same cell line, DHEA was able to phosphorylate BCL2, leading to its dissociation from BECLIN1 and, consequently, inducing autophagy [270]. Although these promising findings highlight the potentiality of DHA and DHEA for anticancer treatment interventions, we will need more in vitro and in vivo studies to better examine their use in anticancer therapies.

#### 3.2.6. Cholesterol

Cholesterol is mainly biosynthesized in the endoplasmic reticulum (ER) of hepatocytes. This process requires acetyl-CoA, which is produced primarily during β-oxidation of FAs in the mitochondria [271]. Cholesterol is one of the major products synthesized by the mevalonate cascade, and is necessary for the formation of cellular membranes, modulation of cell membrane fluidity, cellular structures, and the synthesis of hormones and vitamin D [272]. The rate-limiting step in cholesterol biosynthesis is the conversion of 3-hydroxy-3-methylglutaryl-CoA (HMG-CoA) to mevalonate by HMG-CoA Reductase (HMGCR) [273], which is further processed into cholesterol and other lipid molecules, such as GGPP, FPP, and squalene [274,275]. As discussed previously, the production of lipids, including cholesterol, can be inhibited by the cholesterol-lowering agents known as statins. Statins inhibit the rate-limiting enzyme in this pathway, HMGCR [200].

The current knowledge concerning how cholesterol can modulate autophagy in the context of cancer is very limited. One study has shown that the inhibition of cholesterol biosynthesis by statins in blood cancer cells induces autophagy through the inhibition of mTOR signaling [204]. Autophagy was also induced after the treatment of these cells with methyl-β-cyclodextrin, a cholesterol-reducing agent, indicating that autophagy induction by statins was due to cholesterol depletion. The cholesterol–mTORC1 axis regulated autophagy through Golgi membrane protein 1 (GOLM1) and by interacting with LC3 through an LC3-interacting region (LIR) in hepatocellular cancer cells [276]. The cholesterol of the cytosolic membrane lipid raft regulated the autophagic cell death in breast cancer cell lines (MDM-MB231) via the regulation of caspase-8 activity [277]. We have summarized the impact of lipid metabolites in autophagy process in Table 3. 

## 4. Lung Cancer, Metabolism and Autophagy

Lung cancer is a major contributor to the global cancer burden. It is the second most diagnosed cancer in the world, only recently being surpassed by breast cancer in 2020 [278,279]. Globally, lung cancer accounts for 11.4% of all newly diagnosed cancers, representing 1 in 10 new cases [278,279]. It continues to be the number one cause of cancer-related deaths worldwide, accounting for 1 in 5 cancer deaths in 2020 [280]. The death toll from lung cancer exceeds that of breast, prostate, and colorectal cancer combined [278,280]. The five-year survival rate of lung cancer (18.4%) is lower than colorectal (64.5%), breast (89.6%) and prostate (98.2%) cancers [280]. Metastasis, the dissemination of a primary tumor to other parts of the body, is the major cause of morbidity and mortality in lung cancer. Only 16% of cases of lung cancer are diagnosed at an early stage when the cancer is still localized to the lungs [280]. The five-year survival rate is only 5% for those with distant metastases, compared to 56% for those detected at an early stage [280]. Diagnosis at the advanced stage contributes to the substantial mortality in lung cancer, with more than half of people dying of lung cancer within a year of their diagnosis [280].

In the following sections, we will focus on the impact of metabolism and autophagy in the pathogenesis of lung cancer.

### 4.1. Etiology, Pathogenesis, and Natural History of Lung Cancer

The majority of lung cancers are either non-small cell lung cancer (NSCLC) or small cell lung cancer (SCLC). NSCLC accounts for more than 80% of all lung cancers. Adenocarcinoma and squamous cell cancers are the most common subtypes of NSCLC [281]. Adenocarcinoma is currently the most common histologic subtype worldwide and histopathologically diagnosed through its specific morphology and immunohistochemical profile that includes thyroid transcription factor-1 (TTF-1) and Napsin [282,283,284]. About 5–15% of adenocarcinomas harbor targetable genetic driver mutations including EGFR, ALK, BRAF and KRAS. The signaling proteins in these genes drive the proliferation of the malignant cells [285]. Programmed cell death-ligand 1 (PDLI) is a cell-surface protein expressed in NSCLC patients and is associated with poor prognosis and a therapeutic target for immune checkpoint inhibitors [286,287,288]. Squamous cell carcinoma (LUSC) was previously the most common subtype but is now the second most common [282,289]. The majority of squamous cell carcinomas present as more central lung cancers (i.e., closer to the central bronchial tree). It is often defined by immunohistochemical staining for p40, p63, and CK5 or CK5/6 [282,290].

Although there is a combination of genetic and environmental risk factors for the development of lung cancer, the strongest risk factor is known to be environmental exposure to smoking [291]. Although there are growing populations of non-smoking individuals developing lung cancer, smoking is estimated to be a risk factor in 90% of patients who develop lung cancer [291]. Smoking exposure as a risk factor for developing lung cancer not only applies to people who smoke but also extends to second-hand smoke (involuntary inhalation of smoke by non-smokers) or third-hand smoke (exposure to smoke deposited on surfaces, such as clothing, furniture, etc.) [292,293]. Although the data on third-hand smoke exposure is developing, there is established evidence that second-hand smoke exposure is associated with a higher risk of developing lung cancer. The evidence suggests that there is a dose–response relationship between second-hand smoke exposure and the incidence of lung cancer among never-smokers [292,293]. For example, women who have never smoked had a 27% higher incidence of lung cancer if their spouse smoked [292].

A common established risk factor for lung cancer is exposure to radon [294,295]. Radon is a gas resulting from the decay of radium and uranium [294,295]. Due to its potential presence in soil and groundwater, it can potentially accumulate in homes [294,295]. Asbestos is also a known risk factor for lung cancer and appears to have a synergistic effect with smoking to increase the risk of lung cancer [296,297]. Asbestos exposure can be due to occupational exposure or to exposure in homes or other buildings (i.e., schools, workplaces) [296,297]. Other environmental/occupational exposures associated with an increased risk of lung cancer are chromium, nickel, and arsenic [298]. Pulmonary fibrosis appears to increase the risk of developing lung cancer even after controlling for the effect of smoking [299]. HIV and HPV infection may also increase the risk of developing lung cancer, although it is unclear if this observation is being driven predominantly by smoking exposures [300,301,302,303].

Genetic factors are also important. The most common genetic risk factors are mutations in EGFR (epidermal growth factor receptors) and ALK (anaplastic lymphoma kinase) [304,305]. EGFR mutations are more common in Asian populations [306].

As NSCLC is the most common lung cancer, we will focus this part of our review on this type of lung cancer. We will first explain the genetic mutations of autophagy in this type of the lung cancer. Then, we will discuss how changes in metabolism could affect autophagy in NSCLC.

### 4.2. Modulation of Autophagy through Genetic Changes in NSCLC

Components of the Rho GTPase signaling pathway are found to be significantly mutated in NSCLC. For example, inactivating mutations in the Rho GTPase activating protein (RhoGap) *ARHGAP35* (p190RhoGAP) are found in ~6% of LUSCs, which results in increased RhoA GTPase activity. Autophagy and Rho GTPase signaling show reciprocal regulation [307]. Previously, it was shown that the knockdown of *ROCK1* (Rho-associated kinase 1), a downstream effector of RhoA, was found to enhance starvation-induced autophagy [308]. In addition, increased ROCK signaling inhibited autophagy, whereas the inhibition of ROCK resulted in an augmented autophagy flux with an increased size of early and late autophagic structures [309]. The authors suggested that ROCK1 signaling delimits the elongation phase and the eventual size of newly-formed autophagosomes [309]. Subsequently, it was found that ROCK1 promoted starvation-induced autophagy through the phosphorylation of BECN1 (*ATG6*) to prevent BECN1–BCL2L1 interaction [310]. Sequestosome-1 (p62 SQSTM1), an autophagy receptor, sequesters RhoA for autophagic degradation [311]. In addition, the knockdown of *ATG5* results in enhanced RhoA levels and defective cytokinesis, increasing aneuploidy and motility [311,312]. The knockout (KO) of *Atg5*, *Atg7*, and *Ulk1* in Mouse Embryonic Fibroblasts (MEFs) resulted in increased cell motility with an amoeboid morphology [313]. Tumor cells have two major forms of motility, mesenchymal and amoeboid-type movements, controlled by the Rac1 and RhoA GTPases, respectively [314]. Interaction with p62 SQSTM1 results in degradation of the Rho guanine nucleotide exchange factor *ARHGEF2* (GEF-H1) [313]. *Atg7* KO cells show increased ARHGEF2 and RhoA activity [313]. Lung adenocarcinomas (LUADs) show mutations in the *ARHGEG2* gene (4%); however, the biological consequence of these mutations is not known at this time. Overall, enhanced RhoA activity increases stress-induced autophagy and motility, which, in turn, sets up a negative feedback loop through autophagy-mediated degradation of RhoA and its GTP exchange factor, ARHGEF2. It has been recently reported that circular RNA Rho GTPase activating protein 10 (circARHGAP10) is an important factor for the development of NSCLC [315]. circARHGAP10 has been shown to be upregulated in NSCLC tissues, cells, and serum-derived exosomes. On the other hand, serum-derived exosomes increased circARHGAP10 expression in NSCLC cells [315]. miR-638 is a target of circARHGAP10 and is involved in its effect on proliferation, migration, invasion, and glycolysis in NSCLC cells [315]. Another investigation showed that low expression of ARHGAP6 is linked to increased metabolic activity in NSCLC, which confirms that ARHGAP6 is a potential tumor suppressor in NSCLC [316]. Therefore, it could be concluded that there might be an indirect link between the effect of small Rho GTPase on glycolysis via the regulation of autophagy.

The *BCL2L1* gene is focally amplified in ~8% of lung LUSCs and 3% of LUADs. BCL2L1 binds BECLIN1 [317], inhibiting its ability to activate autophagy [318,319]. Therefore, elevated levels of BCL2L1 would be expected to inhibit autophagy induction [319]. In addition, elevated levels of BCL2L1 would also be anti-apoptotic through the binding of pro-apoptotic proteins, such as BAX and BAK [320]. This inhibition of apoptosis would also impart drug resistance to tumor cells [321]. It is still an open question whether BCL2 and BCL2L1 can alter autophagy directly [322]. One study using *BAX*, *BAK1* knockout cells, to prevent apoptosis induction, showed that overexpression of *BCL2*, *MCL1,* and *BCL2L1* in the knockout cells did not alter autophagy [323,324]. Only when *BAK* and *BAX* were present, was apoptosis-induced autophagy activated. This suggested that BCL2-like proteins regulate autophagy indirectly by inhibiting BAX/BAK1-mediated apoptosis [323,324]. A follow up study, from another group, showed that the BH3 peptidomimetic drug ABT-737 induced autophagy in *BAX*/*BAK1* double knockout cells when longer durations and higher doses were used, which correlated with the disruption of BCL2–BECN1 binding [325]. However, a further study from Reljic et al. suggested that higher doses of BH3 peptidomimetics caused off target effects [326]. Using the cBioPortal.org web site and TCGA data, LUAD patients with *BCL2L1* amplification showed a significant reduction in progression-free survival but not overall survival (Figure 8A,B). However, the number of patients with amplified *BCL2L1* was small (*n* = 7) and more patients with this genotype should be studied [327,328,329]. Further studies are needed to determine whether the effects of *BCL2L1* amplification is due to alterations of apoptosis or autophagy. One of the hallmarks of cancer is the evasion of apoptosis [330,331], which is associated with an increase in invasion and chemo-resistance [332], A part of the decrease in apoptosis in tumor cells is correlated with changes in the balance between the BCL2 family of pro- and anti-apoptotic proteins [333]. Therefore, the process of apoptosis, autophagy, and the metabolic status of cancer cells, including NSCLC tumor cells, are tightly linked to BCL2 family proteins and the genetic changes of this family could have a direct effect on the regulation of tumor cell response to chemotherapy or tumor invasion.

The frequent mutation in PI3 kinase signaling pathway genes in lung tumors results in activation of the mTORC I kinase complex, which, in turn, would inhibit autophagy (for review [334]). This suppression of autophagy by oncogenes or tumor suppressors may cause an acceleration of early-stage tumor formation. In a *K-ras^G12D^*-induced mouse model of NSCLC, it was shown that the knockout of *Atg5* resulted in increased tumor growth during the early stages compared with autophagy competent mice [335]. The accelerated early oncogenesis was due to increased infiltration of FOXP3^+^ regulatory T cells, which subverted immunosurveillance [335]. However, the *Atg5* knockout mice showed increased overall survival due to the reduced progression of adenomas to adenocarcinomas, which could be overcome by *Tp53* loss [335]. Another study using *Atg7* knockout mice showed that autophagy was needed for sustained K-ras-induced tumor cell proliferation and progression [205]. The lack of *Atg7* resulted in oncocytoma formation due to the accumulation of morphologically abnormal mitochondria [205]. *Tp53* loss allowed the progression to adenocarcinomas in this study; however, the acceleration of early oncogenesis was not seen [205]. It was previously shown that wild type *TP53* under drug or stress treatments inhibits mTOR1 kinase complex activity and, in turn, activates autophagy [336]. In tumors with *TP53* loss, mTOR1 would not be inhibited resulting in reduced stress induced autophagy. Therefore, the inhibition of autophagy may be another tumor suppressor function of *TP53* [336].

### 4.3. Autophagy, Glucose Metabolism, and Lung Cancer

As we have discussed in previous sections, glucose metabolism is tightly linked to autophagy in different types of cancers. In this section, we focus the discussion on the impact of glucose metabolism and autophagy in lung cancer.

LKB1 is a master tumor suppressor that limits cell growth through repression of the mammalian target of rapamycin complex 1 (mTORC1) and activation of AMPK to control metabolism. Therefore, the loss of LKB1 leads to uncontrolled mTORC1 activation, activation of HIF-1α and, consequently, angiogenesis and the Warburg effect [337,338,339]. Co-mutations in the KRAS proto-oncogene and the LKB1 tumor suppressor gene frequently occur in hyper-metabolic and aggressive human lung adenocarcinoma tumors. mTORC1 is hyperactive in LKB1/KRAS mutant lung adenocarcinoma cancer cells and, thus, targeting this pathway by mTOR inhibitors might represent an effective way of treating lung adenocarcinoma [340,341]. In 2018, researchers were able to use the coupling of [18F]-2-fluoro-2-deoxy-D-glucose positron emission tomography (18F-FDG PET)/computed tomography (CT) imaging to quantitative immunohistochemistry (qIHC) as an effective method to measure metabolic changes and therapeutic responses caused by a selective catalytic mTOR kinase inhibitor in mouse models of lung cancer. Their analysis showed that the mTOR inhibitor MLN0128 could suppress tumor growth and glycolysis in genetically engineered mouse models (GEMMs) of lung cancer with KRAS and LKB1 co-mutations, as shown by reduced 18F-FDG consumption [342]. Interestingly, in addition to the AMPK-mTOR signaling pathway, SIRT3/ HIF-1α was significantly upregulated in KRAS mutant human lung adenocarcinoma lines (H460, A549, and H358) [343]. Lian-Xiang Luo demonstrated that the simultaneous use of an antitumor (Honokiol) and an autophagy inhibitor, 3-methyladenine (3-MA) triggered pro-death autophagy through activating mitochondrial SIRT3, suppressing HIF-1α expression, and disrupting oncogenic KRAS-mediated RAF/MEK/ERK and PI3K/AKT/mTOR signaling pathways [343]. The last step of glycolysis is the conversion of PEP to pyruvate and ATP, which is catalyzed by PK. In KRAS mutant cells, PKM2 is hyperactivated and acts as a histone kinase in the nucleus and upregulates the expression of c-Myc. Their overexpression affects metabolic functions including the enhanced expression of glucose transporter GLUT1, the activity of glycolytic enzymes (HK2, PFK, enolase1), and LDHA, with the overproduction of lactic acid and non-metabolic pathways (upregulation of autophagy). Even though PKM2 activates the mTOR pathway resulting in a block in autophagy in KRAS mutants. Autophagy is hyperactive in KRAS mutant cancer cells [344]. M Morita used KRAS in PKM2 mutant lung cancer mouse models (Kras^G12D^ or Kras^G12V^) and in SCLC cells [345]. In addition to KRAS, oncogenic promoters such as the EGFR support glycolytic metabolism in lung tumors. EGFR signaling activates the class I PI3K-AKT-mTORC1 pathway. AKT and mTORC1 phosphorylate BECLIN1 on serine residues, leading to downregulation of VPS34 activity and autophagy. In contrast, ULK1 and AMPK stimulate autophagy by phosphorylating BECLIN1 and promoting the formation of the active BECLIN1–VPS34 complex. mTORC1 also suppresses autophagy initiation by phosphorylating and inhibiting ULK1. These EGFR-dependent events are blocked by tyrosine kinase inhibitors (TKIs) [346]. Therefore, it can be concluded that autophagy plays an important role in KRAS and EGFR mutant lung tumors through the small Rho GTPase and PI3-AKT-mTORC1 pathway.

M Ye et al. showed that overexpression of GLUT1 and MCT-4 led to a significant increase in glycolysis and autophagy in ER6 cells (erlotinib-resistant sub-line of HCC827) compared to EGFR-TKI sensitive HCC827 lung adenocarcinoma cell lines. They demonstrated that the combination of an AKT inhibitor (MK2206), to suppress AKT phosphorylation or autophagy, with glucose deprivation and a GLUT1 specific inhibitor (STF-31) could tackle the resistance to erlotinib in NSCLC [347]. Other compounds that target the EGFR are flavonoids (natural products with variable phenolic structures that reduce the risk of cancer). A recent study revealed that Scandenolone (a natural isoflavonoid) directly attaches to the ATP-binding site of EGFR to impede the phosphorylation of the AKT/ERK signaling pathway, which is essential for sensitizing SK-MEL-28 melanoma cell lines to intrinsic apoptosis and blocking autophagy flux. In fact, exposure to Scandenolone inhibits the AKT/ERK pathway, leading to an increase in Bad, one of the BH3-only proteins, and a reduction in Bcl-2, an anti-apoptotic protein, disrupting not only the inhibitory interactions between BECLIN1 and Bcl-2 but also the activation of the lysosomal protease Cathepsin B, which induces the induction of autophagy in these cells [347]. Another study examined whether adding an additional isopentenyl group to the Scandenolone structure would improve its anticancer effect. This new antitumor compound, named Final-2, could not only suppress cancer cell growth by promoting apoptosis and blocking autophagy but also by inhibiting the expression of the oncogenes KRAS, c-MYC, and glucose transporters (GLUTs), such as GLUT1, GLUT3, and GLUT4, which resulted in the downregulation of glucose metabolism in the human lung cancer cells A549 [348].

It has been shown that under stress conditions (hypoxia, glucose deprivation, low pH, ROS accumulation, and Ca^2+^ homeostasis perturbation), lung cancer cells experience ER stress and activate GRP78/BiP-related unfolded protein response (UPR) signaling. Upon ER stress, GRP78/BiP can trigger autophagy to improve the survival of cancer cells through three different UPR pathways including eukaryotic translation initiation factor 2 alpha kinase 3/PKR-like ER kinase (EIF2AK3/PERK), endoplasmic reticulum to nucleus signaling 1/inositol-requiring enzyme 1 (ERN1/IRE1), and activating transcription factor 6 (ATF6), which target the ATG16L1 complex, MAPK/JNK and X-box binding protein 1 (XBP1), and the AKT-mTOR pathway, respectively [349,350,351,352]. It has been reported that overexpression of GRP78 is often seen in advanced-stage lung cancer, which is associated with a poor prognosis [353]. A recent study has reported that the inhibition of ATPase activity of GRP78 by a thiazole benzenesulfonamid (HA15) provokes ER stress-mediated UPR and induces pro-apoptotic autophagy in the lung cancer cell line A549 [354]. Glucose starvation induces ER stress-related GPR78 in NSCLC cells, leading to autophagy as a cyto-protective process. In this regard, the microtubule cytoskeleton is involved in autophagy signaling and βIII-Tubulin is one of the β-tubulin isotypes overexpressed in glucose-starved NSCLC cells. On the other hand, GRP78 interacts with βIII-tubulin; however, the functional consequences of this association are unknown. Tubulins, and in particular βIII-tubulin, associate with a variety of glycolytic enzymes including PK, phosphofructokinase, aldolase, hexokinase, GAPDH, and LDH. These observations suggest an inherent link between the microtubule cytoskeleton, glycolysis, autophagy, and the initiation of ER stress responses [355]. βIII-Tubulin enables rapid GPR78/Akt activation in response to glucose starvation and decreases the reliance of cells on glycolytic metabolism to maintain cell survival and proliferation. Amelia L. Parker, in 2016, indicated that suppression of high βIII-tubulin expression with 4-phenylbutyric acid and N-acetylglucosamine ameliorates ER stress and reduces autophagy by delaying the association of GRP78 with Akt in glucose-starved NSCLC cells [356]. The major stressors influencing autophagy flux are Ca^2+^ homeostasis disturbances and glucose starvation. Thus, voltage and ligand-gated calcium channels (such as T-type, L-type, and IP3-R), and GLUTs, which are often upregulated in cancer, are also targets for autophagy regulation. Both Ca^2+^ and glucose influx perturbations might be signals for UPR and ER stress activation, leading to the ER chaperones such as GRP78/BiP stimulating autophagy through three different molecular pathways, such as ERN1/IRE1-JNK-Bcl-2, EIF2AK3/PERK-eIF2a-ATF4, or ATF6-XBP1-ATG. In addition, both starvation and Ca^2+^ perturbations may lead to activation of ER stress-mediated autophagy in cancer cells via distinct Ca^2+^ channel activation, CaMKK*β*, which mediates AMPK-dependent inhibition of mTORC1 or by activating the IP3-R-BECLIN1-Bcl-2 pathway. Therefore, the development of Ca^2+^ channel and GLUT blockers would be a novel cancer therapy. Hong-Kun Rim et al. revealed that co-blocking the T-type Ca^2+^ channel and GLUT with KYS05047 and KYS05090 synthesized compounds can induce autophagy and apoptosis in human lung adenocarcinoma A549 cells and xenografts resulting in elevated ROS generation and reduced intracellular Ca^2+^ levels and glucose uptake [352,357,358,359].

Glutamine, the blood’s most abundant amino acid, is metabolized through glutaminolysis, catalyzed by glutaminase (GLS) and leucine-dependent glutamate dehydrogenase (GDH), to produce αKG for the Krebs cycle, nucleotides, and other amino acids. The expression level of GLS is regulated by the oncogene c-MYC, and its upregulation is a hallmark of tumor growth. In fact, enhancing glutaminolytic αKG production stimulates lysosomal translocation and the activation of mTORC1. mTORC1 (master switch for cell proliferation and autophagy) is regulated by glutaminolysis [360]. Recently, RC Bruntz et al., by using [^13^C6]-glucose and [^13^C5,^15^N2]-glutamine as tracers, showed that Selenite inhibited glutaminolysis by suppressing GLS1 expression and inducing excess ROS production, which mediates autophagy and eventual cell death in NSCLC A549 cells [361]. The upregulated glutaminolysis is a feature of cancer cell metabolism, activated by extracellular lactate signaling. Under oxygenated conditions in the tumor environment, lactate enters the cells via MCT1, stabilizing hypoxia-inducible factor-2α (HIF-2α), and then transactivating c-MYC. This, in turn, activates the expression of glutamine transporters (LAT1 and ASCT2) and glutaminase 1 (GLS1), resulting in an increased proportion of glutamine absorption by tumor cells and triggering OXPHOS [362,363]. Forming an acidic tumor microenvironment is a common event in lung cancer due to the production of lactate by the Warburg effect. For lung cancer cells to survive in such a situation in vivo or in vitro, autophagy is induced by the acidic pH through ER stress-related GRP78. Glucose starvation and the inhibition of MCT-1 with a-cyano-4-hydroxycinnamate (CHC) in mice injected with hypoxic Lewis lung carcinoma (LLC) cells resulted in decreased tumor growth [364]. The knockdown of GRP78 by siRNA reduced autophagy activation under acidic conditions in the human lung cancer cell lines (A549 and NCI-H226) and enhanced acid-induced apoptosis [365]. Glucose depletion (GD) induces necrotic cell death in A549 lung carcinoma cells, activating an inflammatory response due to the release of high-mobility group box 1 (HMGB1). SC Lim et al. revealed that sodium salicylate, an active metabolite of aspirin, could inhibit HMGB1 release, Cu/Zn superoxide dismutase1 (SOD1) release, and ROS production in A549 cells cultured in GD medium, which, in turn, led to the anti-inflammatory effect. Moreover, aspirin could activate AMPK, inhibit mTOR to induce autophagic cell death, and suppress tumor development [366,367,368]. Glutamate oxaloacetate transaminase 1 (GOT1), by catalyzing the conversion of aspartate to oxaloacetate (OAA), provides the metabolites necessary for gluconeogenesis and, thus, GOT1 is involved in the maintenance of redox homeostasis and the regulation of glycolytic metabolism in normal cells [369], while the inhibition of GOT1 leads to increased glucose utilization and lactate production, resulting in a decrease in cellular pH [370]. X Zhou in 2018 demonstrated that in GOT1-null 143B osteosarcoma cells or GOT1 siRNA knockdown in A549 lung cancer cells resulted in the elevated expression of autophagy-related genes (ATG5 and Beclin1) and increased secretion rate of lactate compared with wild type cells [370]. Placing these cells in a glucose-free culture medium caused a reduction in cell viability [370].

Nickel compounds are mainly associated with metabolic changes in sino-nasal and lung cancers. YT Kang et al., in 2017, documented that hexokinase 2 (HK2), involved in the first step of glycolysis, plays a significant role in Nickel-induced autophagy in lung bronchial epithelial cells (BEAS-2B). They determined that the inhibition of HK2 by the antihyperglycemic drug metformin and the knockdown of lipocalin 2 (LCN2) by shRNA mitigated NiCl2-mediated autophagy and induced apoptosis in BEAS-2B cells [371]. VK Chaudhri et al., by evaluating two cell groups including primary human lung tumor CAFs and “normal” fibroblasts (NF) isolated from nonneoplastic lung tissue located at least 5 cm away from the tumor, determined that metabolic alterations are not limited to cancer cells. They demonstrated that the reduction in the amount of glucose in the medium from 25 to 5 mmol/L also increased dipeptides and autophagy in NFs [372].

We have summarized the role of various glycolytic enzymes in relation to autophagy in lung cancer in Table 4.

### 4.4. Lipid Metabolism Regulation of Autophagy in Lung Cancer

#### 4.4.1. HMGCR Inhibition

Statins (inhibitors of HMGCR) are widely used for lowering serum cholesterol. They also have several pleiotropic effects including their potential effects in different types of cancers [201,381]. A recent xenograph study in nude mice by Yang et al. reported that the use of the statin fluvastatin could significantly inhibit the metastasis of lung adenocarcinoma cells to bone. The anti-metastatic effect of fluvastatin was shown to be due to the induction of autophagy in lung adenocarcinoma cells. The chemical inhibition of autophagy by 3-methyadenine (3-MA) or Bafilomycin A1 (Baf A1) and the genetic inhibition of autophagy by knocking down *Atg5* or *Atg7* hampered the suppressive effects of fluvastatin on bone metastasis of lung cancer cells. These findings highlight the importance of the mevalonate pathway inhibition by statins (through HMGCR inhibition) on the regulation of autophagy and how this leads to the suppression of bone metastasis of lung cancer cells [382].

Another statin that has been extensively studied in cancer is lovastatin. It has been reported that lovastatin has inhibitory effects on the growth and proliferation of NSCLC adenocarcinoma lung cancer cells A549 [383]. In addition, it increases the sensitivity of A549 lung cancer cells to ionizing radiation and inhibits their colony forming ability. Interestingly, these effects by lovastatin were reversed when mevalonate was used in the treatment as well [383]. This indicates that the effects of lovastatin were dependent on its ability to inhibit the mevalonate pathway. In this same study, the treatment of A549 cells with lovastatin resulted in the induction of cell death and induced AMPK phosphorylation, an upstream regulator of autophagy, by directly activating ULK1 through phosphorylation [383]. Collectively, these results show that the lovastatin-induced cell death could be due to AMPK-mediated lethal autophagy. These results emphasize the crucial role that the mevalonate pathway can play on the regulation of autophagy, which, consequently, impacts lung cancer cells’ proliferation and clonogenicity and improves their sensitivity to ionizing radiation [382].

#### 4.4.2. SPHK1

The expression of SPHK1 is significantly increased in NSCLC patients. In a study by Zhu et al. [384], it was found that SPHK1 expression is highly elevated in A549 cells. They showed that SPHK1 overexpression is enhanced in A549 cells while SPHK1 knockdown prevented the migration and invasion of these cells. In fact, the expression of E-cadherin was reduced, and the expression of SNAIL was increased in overexpressed cells. More importantly, SPHK1 induced AKT activation. Since the mTOR-AKT pathway is a master regulator of autophagy, it is safe to say that SPHK1 promotes the invasion and migration of lung cancer cells simultaneously with (and possibly through) autophagy inhibition [384]. This evidence highlights that the targeting of SPHK1 is a promising candidate for NSCLC treatment. Consistent with these findings, another research group evaluated the expression levels of SPHK1 in tissues obtained from 176 NSCLC patients using immunohistochemistry (IHC) [385]. The expression of SPHK1 was correlated with clinicopathological factors and it was shown that the patients with a high expression of SPHK1 that were treated with adjuvant platinum-based chemotherapy had a significantly lower survival rate. This indicates that even performing IHC on tissues from NSCLC patients can be a potential predictive tool for the survival of NSCLC patients.

In addition, another study used SPHK1 expression levels to predict survival and progression in NSCLC patients. Findings from this study revealed that SPHK1 expression was significantly increased in 218 tissue samples from NSCLC patients [386]. This was associated with an inhibition of doxorubicin- or docetaxel-induced apoptosis and the induction of anti-apoptotic factors. Conversely, the chemical inhibition and knockdown of SPHK1 remarkably increased NSCLC cells’ sensitivity to apoptosis triggered by anticancer agents both in vitro and in vivo. Regarding the association of SPHK1 and autophagy, it was also shown that high SPHK1 expression induced the activation of the PI3K/AKT pathway, which, as mentioned earlier, can inhibit autophagy via mTOR signaling. Interestingly, inhibition of the PI3K-AKT pathway failed to induce anti-apoptotic effects by SPHK1; therefore, highlighting this pathway and its downstream pathway (autophagy) as a mediator of SPHK1 effects on NSCLC cells. In agreement with these findings, it was found that SK1-I, a competitive SPHK1 inhibitor, induces apoptosis in cancer cells and suppresses the metastasis of cancer cells into lymph nodes [387]. As mentioned in previous sections, phosphoinositides can modulate autophagy. In addition, any deregulation in the function of PI3K leads to autophagy deregulation because PI3K is part of the PI3K-AKT-mTOR pathway [79].

#### 4.4.3. PA

Phospholipase D (PLD) is the enzyme that hydrolyses phosphatidylcholine (PC) to PA and choline [388]. The expression and activity of a single nucleotide polymorphism (SNP) in the Phospholipase D1 (PLD1) gene (SNP A2698C) is significantly increased in a variety of human cancers [389]. It has been shown that PLD1 expression is significantly increased in lung surgical specimens obtained from NSCLC patients compared to normal lung tissues [390]. Moreover, PLD1 has been associated with the development and aggression of lung cancer [386].

As discussed in earlier sections, PA is involved in (1) the formation of autophagosome curvature, and (2) the inhibition of mTOR and, therefore, the induction of autophagy [81,391]. It can be suggested that the higher expression of PLD1 in NSCLC leads to the higher production of PA and, consequently, higher rates of autophagy induction in NSCLC cells [389]. In addition, it was reported some time ago that upon activation (through bradykinin and sphingosine 1), PLD1 is involved in the PKC pathway in NSCLC A549 cells [392,393], which can regulate autophagy. This further suggests the regulation of autophagy by lipid metabolism in NSCLC.

Lipin-1 is the enzyme that converts PA to diacylglycerol (DAG) [394]. Lipin-1 expression is significantly increased in NSCLC cell lines and patient tissues. It has recently been shown that the elevated expression of Lipin-1 is associated with poor prognosis of NSCLC patients. Moreover, Lipin-1 suppression greatly reduces the growth and viability of NSCLC cells while it has almost no significant effects on non-cancerous cells in the lung [395]. In addition, the same research group demonstrated the loss of Lipin-1 sensitized NSCLC cells to cisplatin. Since PA is involved in the formation of autophagosome curvature [388], the disruption of Lipin-1 can potentially help accumulate the PA levels and consequently boost autophagosome formation during autophagy activation. This can potentially reveal the impact of PA in autophagy regulation during NSCLC progression. These findings suggest that targeting Lipin-1 (and, therefore, the conversion rate of PA to DAG) in combination with other NSCLC anticancer agents could be a promising treatment intervention for lung cancer [395].

Interestingly, it has been observed that the levels of Interleukin-8 (IL-8 or CXCL8), which is a pro-inflammatory chemokine, are significantly elevated in different cancers including NSCLC. This increased IL-8 level has been associated with the enhanced activity of PLD and the activation of AKT/mTOR, which can directly regulate autophagy [389]. Furthermore, IL-8 activates PLD, which consequently converts PC to PA. These findings similarly show the involvement of PDL (and hence PA) in the regulation of autophagy in lung cancer cells.

Due to their importance in cancer development and progression, targeting PDL1 and its signaling molecules has been suggested as a treatment strategy in different cancers with an increased PLD1 activity and expression [388].

#### 4.4.4. Peroxisome Proliferator-Activating Receptors (PPARs)

As mentioned previously, PPARs are ligand-dependent nuclear transcription factors. In order to be functional as transcription factors, PPARs need to bind to lipid ligands (e.g., oleic acid, linoleic acids, linolenic acids, prostaglandins, eicosanoids, and oxidized lipids) through lipid-binding proteins [244,245,246]. A study [396] showed that the treatment of A549 NSCLC cells with troglitazone (an activator of PPARγ) induced the conversion of LC3-I to LC3-II and increased the degradation of p62, indicating activation of autophagy flux while inducing apoptosis in A549 cells. Consistently, the activation of PPARγ and, subsequently, autophagy activation was inhibited when the cells were treated with a PPARγ antagonist, GW9662. This highlights troglitazone as a potential candidate for the combination therapy of apoptosis-resistant NSCLC. Moreover, another study on human NSCLC cell lines, HCC827 and H1650, demonstrated that PPARγ agonists sensitize PTEN-deficient lung cancer cells to EGFR TKIs via autophagy induction [397]. One of the mechanisms of developing resistance in lung cancer is through PTEN loss, which contributes to the resistance to EGFR TKIs [398]. It was shown that PPARγ agonists increased the expression of PTEN and, consequently, inhibited the PI3K-AKT signaling pathway, which will activate autophagy through mTOR inhibition. This results in significant sensitivity to gefitinib (an EGFR TKI). The mechanism of this sensitivity to EGFR-TKIs was found to be the induction of autophagy. In fact, the findings by this group showed that PPARγ agonists induced higher levels of LC3-II accumulation and the degradation of p62 in NSCLC cells. In addition, genetic silencing of *ATG5* eliminated this potentiation effect by PPARγ agonists. Therefore, the combined use of PPARγ agonists and EGFR TKIs have recently been suggested for the treatment of lung cancer patients. In an animal study [399], it was shown that the treatment of mice with pioglitazone (synthetic PPARγ ligand) remarkably prevented lung tumor formation induced by 4-(methylnitrosamino)-l-(3-pyridyl)-lbutanone (NNK). In addition, it was found that endogenous PPARγ was decreased well before lung tumor formation, indicating a role for the endogenous PPARγ molecular pathway in lung cancer development. These findings show that the potential increase in the activity of PPARγ using its ligands not only could be inhibitory for lung tumor formation but also beneficial for the treatment of lung cancer. Moreover, assessment of the activity levels of PPARγ may be useful as biomarkers for lung cancer.

#### 4.4.5. Omega-3 Polyunsaturated Fatty Acids (n-3 PUFAs) (Docosahexaenoic Acid (DHA), Docosahexaenoyl Ethanolamine (DHEA), EPA)

Free lipids can also regulate autophagy during lung cancer, although only a handful of studies have investigated this relationship. n-3 PUFAs and, in particular, DHA or EPA, have been defined as important preventive lipids against lung cancer development [400,401]. They can exert their effects through different ways including cell surface receptor functions, membrane fluidity, and the enhancement of oxidative stress, all of which might induce autophagy [401]. One study [402] demonstrated that both DHA and EPA caused the formation of autophagosomes in a concentration-dependent fashion in the NSCLC A549 cell line. This was confirmed by the finding that treatment of A549 cells with the autophagy inhibitor, 3-MA, reversed the formation of autophagosomes. This autophagy inhibition also prevented apoptosis (reduced activation of caspase-3/7) and increased cell viability, showing that DHA and EPA-induced autophagy acts to enhance the apoptotic cell death of A549 cells. It has also been reported that the treatment of A549 cells with DHA or EPA induced autophagy in a dose- and time-dependent manner [262]. The results from this study also showed that the anti-tumor effects of DHA and EPA were associated with the activation of autophagy.

## 5. Conclusions and Future Directions

Autophagy plays an important role at different stages of a tumor’s development, progression, and metastasis. During the early stages of tumor development, it prevents tumorigenesis by the removal of damaged mitochondria and misfolded proteins, while it supports tumor progression in the later stages of tumor formation. Autophagy also regulates tumor metastasis both positively or negatively; however, it is dependent on the type of tumor. As we have discussed in our article, both glucose and lipid metabolism can regulate autophagy via different mechanisms to compensate for the high ATP demand of cancer cells, as summarized in Figure 9. The enzymes of glycolysis can directly interact with the autophagy machinery (HK2) [107] to change their activity or expression levels or to directly induce changes in autophagy in tumor cells (PFKFB4) as an adaption to cellular stress conditions [124,125]. On the other hand, the major adaptations of lipid metabolism observed in cancer are generally associated with changing the autophagy pathway in favor of the increased aggressiveness of cancer cells. Therefore, targeting lipid metabolism or glycolysis is an attractive therapeutic strategy.

The interconnection between lipid/carbohydrate metabolism and autophagy and also the role of these metabolic pathways in the regulation of autophagy could be harnessed to develop new clinical interventions against cancer. There have been multiple approaches, including the development of small molecule inhibitors of enzymes that catalyze ceramide catabolism, synthetic ceramide analogs, inhibitors of sphingosine kinase, and antagonists of the S1P receptor [403,404]. Other approaches also exist; for instance (a) reactivation of the SMase, SPL, and S1P phosphatase genes that are suppressed in cancer cells; (b) targeting and hence inhibiting the dihydroceramide desaturase; and (c) using small molecules to activate SMase enzymes [405]. A study by Hernandez-Tiedra et al. [406] showed that the treatment of U87 glioma cells with Delta 9-tetrahydrocannabinol (THC, the main active component of *Cannabis sativa*) enhances the ratio of dihydroceramide:ceramide in the endoplasmic reticulum of these cells. This ratio change has an impact on the autophagosomes and autophagolysosomes leading to the permeabilization of lysosomal membranes, and the release of hydrolases and cathepsins. Subsequently, this will result in the activation of apoptosis in these cells. Acid ceramidase (AC) is a lysosomal hydrolase that catalyzes the conversion of ceramide into fatty acid and sphingosine. AC has been reported to regulate the levels of molecules that have pro- or anti-apoptotic effects [407,408]. Tumors with high levels of AC expression and, thus, lower ceramide levels, demonstrate higher protection from stress stimuli such as chemotherapy. An innovative approach to avoid and circumvent cancer resistance to single and combined chemotherapy, known as multidrug resistance (MDR), is, thus, the combined use of chemotherapeutic agents and AC inhibitors to better disarm cancer cells [409]. As an example, targeting lipid metabolism to control autophagy in cancer has been widely investigated. One of the important regulators of fatty acid composition in the cell is stearoyl-CoA desaturase 1 (SCD1), which is involved in monounsaturated fatty acid (MUFA) synthesis from saturated fatty acid (SFA) [410]. Furthermore, a connection between SCD1 and autophagy has been reported. In fact, ULK1 inhibition reduces SCD1 expression in liver cells, increasing the SFA/MUFA ratio and, subsequently, leading to cell death [411]. In addition, the autophagy of lipid molecules (lipophagy) provides the necessary fatty acids that are essential for appropriate mitochondrial respiration [412]. Given the lipid metabolism and autophagy crosstalk, and the key role of SCD1 in lipid metabolism, SCD1 is a promising target for developing novel anti-cancer treatments. SCD1 inhibitors, including A939572, CAY10566, MF-438 and CVT-11127, significantly decrease the proliferation rate and induce apoptosis in different cancer types including lung cancer [413,414,415]. That said, due to the adverse effects, most of the studies have not progressed into clinical trials after the pre-clinical studies. Furthermore, due to the SCD1 and autophagy interconnection, new combination therapies with both SCD1 inhibitors and autophagy inducers/inhibitors have been suggested [416,417]. It is, however, worth mentioning that the most effective cancer therapy should be designed based on the biological context; for example, the presence of mutations, tissue type, cancer heterogeneity, etc.

On the other hand, the alteration of various mitophagy modulators has been shown to play a role in tumor progression (for review [418]). LUAD lung cancer patients with a *BNIP3L* deletion did not show significant differences in overall or progression-free survival. BNIP3L acts as an adaptor molecule that targets mitochondria for turnover and recruits components of the mitophagy machinery to the mitochondria [419]. Another study found that 26.7% of lung tumor samples showed reduced expression of BNIP3L but no rearrangement or mutation of the gene was seen [420]. The closely related *BNIP3* had been shown to act like a tumor suppressor in triple negative breast cancers [421,422]. When deficient, it results in increased metastasis and an increase in damaged mitochondrial mass, resulting in the upregulation of *HIF1α* and reactive oxygen species (ROS) levels [421,422].

The *PRKN* (*PARK2* or *PARKIN*) gene is a site of frequent loss of heterozygosity in NSCLCs [423]. Transfection of the *PRKN* gene into a deficient lung cancer cell line reduced in vivo tumor growth but had no effect on in vitro cell growth [423]. Lung cancer mutated forms of PRKN show impaired mitophagy and reduced cell death after mitophagy induction [424]. PINK1 phosphorylation derepresses the PRKN E3 ubiquitin ligase resulting in the ubiquitination of mitochondrial substrates, committing mitochondria to mitophagic turnover (for review [425]). LUSC patients with a *PRKN* mutation or homozygous deletions (7%) showed significantly reduced progression-free survival. Why LUSCs and not LUADs show this reduced progression-free survival when mitophagy genes are deleted is not known at this time but further study is warranted. In breast cancer, PRKN targets HIF-1α for ubiquitination, controlling its stability and resulting in metastasis suppression [426]. Hence, more work is needed to determine the contributions of mitophagy and HIF-1α stability to PRKN’s tumor suppressor function.

Therefore, a greater understanding of the specific roles played by lipid and glycolytic metabolites and/or the enzymes involved in lipid metabolism and glycolysis in cancer, and understanding the role of mitophagy in cancer, is still needed. Our current knowledge of the role that these pathways can play in the regulation of autophagy in different cancers is very much limited to a few studies. Future studies will, undoubtedly, provide us with promising effective candidates for the prevention and treatment of different cancers.

## Figures and Tables

**Figure 1 cancers-15-02195-f001:**
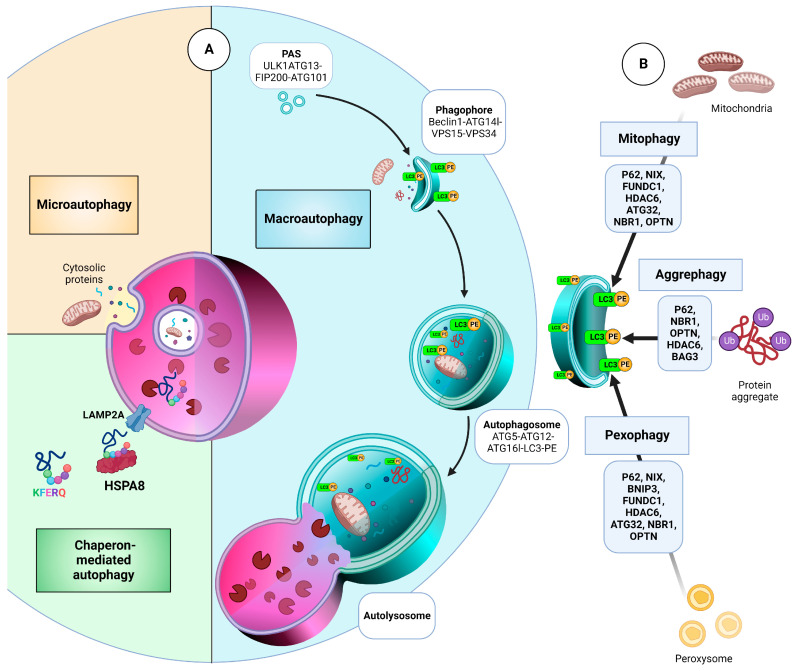
Autophagy is a self-eating mechanism occurring in cells. (**A**) Different steps in the autophagy pathway. Cargos are first engulfed by autophagosomes, which then fuse with lysosomes. Lysosomal enzymes degrade the cargo resulting in the recycling of the cargo’s building blocks. (**B**) Selective autophagy. Autophagy can selectively target and degrade specific cargo using different receptor proteins.

**Figure 2 cancers-15-02195-f002:**
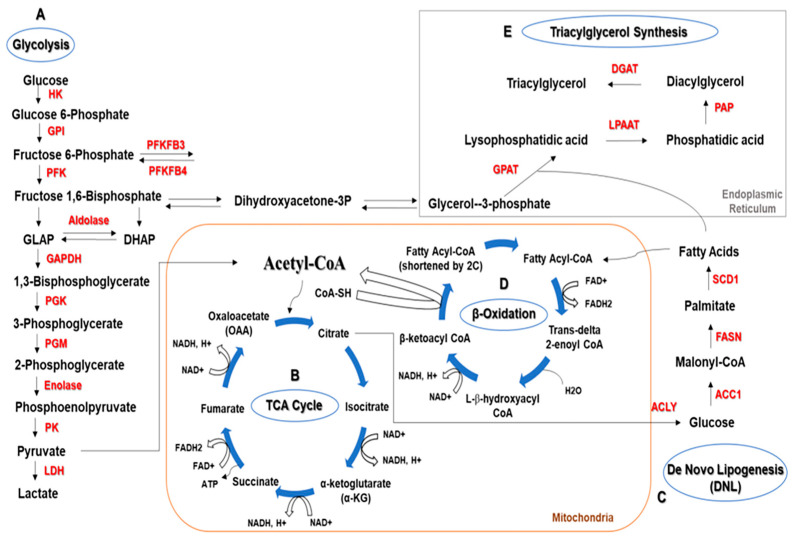
(**A**) **Glycolysis**: This includes 10 consecutive enzymatic reactions in the conversion of glucose into two molecules of pyruvate, which connect with other metabolic pathways. Glucose is phosphorylated to glucose 6-phosphate by the enzyme hexokinase (HK). Glucose 6-phosphate is turned into fructose 6-phosphate by glucose phosphate isomerase (GPI). Phosphofructokinase-1 (PFK) catalyzes the conversion of fructose 6-phosphate and ATP to fructose 1,6-bisphosphate and ADP. 6-phosphofructo2-kinase/fructose 2,6-bisphosphatase (PFKFB), a family of bifunctional enzymes that control the levels of fructose 2,6-bisphosphate (PFKFB3 and PFKFB4 are the two main isoenzymes of this family) (PFK and PFKFB are the two major regulatory enzymes in glycolysis). Subsequently, the aldolase enzyme catalyzes a reversible reaction in which fructose 1,6-bisphosphate converts into glyceraldehyde-3-phosphate (GLAP) and dihydroxyacetone phosphate (DHAP). Glyceraldehyde-3-phosphate dehydrogenase (GAPDH) produces 1,3-bisphosphoglycerate. Next, phosphoglycerate kinase (PGK) forms ATP and 3-phosphoglycerate. Phosphoglycerate mutase (PGM) isomerizes 3-phosphoglycerate into 2-phosphoglycerate. Next, enolase converts 2-phosphoglycerate to phosphoenolpyruvate, which is phosphorylated to form a molecule of pyruvate and a molecule of ATP via pyruvate kinase (PK). Finally, lactate dehydrogenase (LDH) catalyzes the conversion of pyruvate to lactate. (**B**) **Krebs cycle**: The TCA cycle starts with the combination of acetyl-CoA, generated from fatty acids, amino acids, or pyruvate oxidation, with oxaloacetate (OAA) to produce citrate. Citrate is converted into isocitrate. The cycle continues with two consecutive oxidative decarboxylations in which isocitrate is converted into the α-ketoglutarate (α-KG) and the succinyl-CoA concomitantly, producing two CO_2_ and two NADH. Succinyl-CoA coverts into succinate, coupled with the release of ATP. Subsequently, fumarate is formed by the oxidation of succinate. During this step, two hydrogen atoms are transferred to FAD, producing 2FADH. Next, fumarate is converted into malate and further into OAA, which combines with another acetyl-CoA, continuing the cycle [51]. (**C**) **De novo lipogenesis C**: In this simplified scheme, pyruvate from glycolysis feeds acetyl-CoA to the TCA cycle. Citrate from the TCA cycle is converted to acetyl-CoA in the cytosol by ATP citrate lyase (ACLY). Acetyl-CoA from the TCA cycle and other sources is subsequently converted to complicated fatty acids by a series of enzymes including ATP-citrate lyase (ACLY), acetyl-CoA carboxylases 1 (ACC1), fatty acid synthase (FASN), and stearoyl-CoA desaturase-1 (SCD1). (**D**) Fatty acid β-oxidation: Each cycle leads to the formation of acetyl-CoA, nicotinamide adenine dinucleotide (NADH), and flavin adenine dinucleotide (FADH 2). The electron carriers NADH and FADH2 are used by the mitochondrial respiratory chain to generate ATP. (**E**) Triacylglycerol synthesis: This reaction occurs at the surface of the endoplasmic reticulum (ER) bilayer membrane. The first step in this pathway is the acylation of glycerol-3-phosphate by glycerol-3-phosphate acyltransferase (GPAT), producing lysophosphatidic acid. This is followed by further acylation by LPA acyltransferase (LPAAT) and dephosphorylation by phosphatidic acid phosphorylase (PAP) to yield diacylglycerol (DAG). The final step is converting 1,2-diacylglycerol into triacylglycerol (TAG), which is catalyzed by diacylglycerol acyltransferase (DGAT).

**Figure 3 cancers-15-02195-f003:**
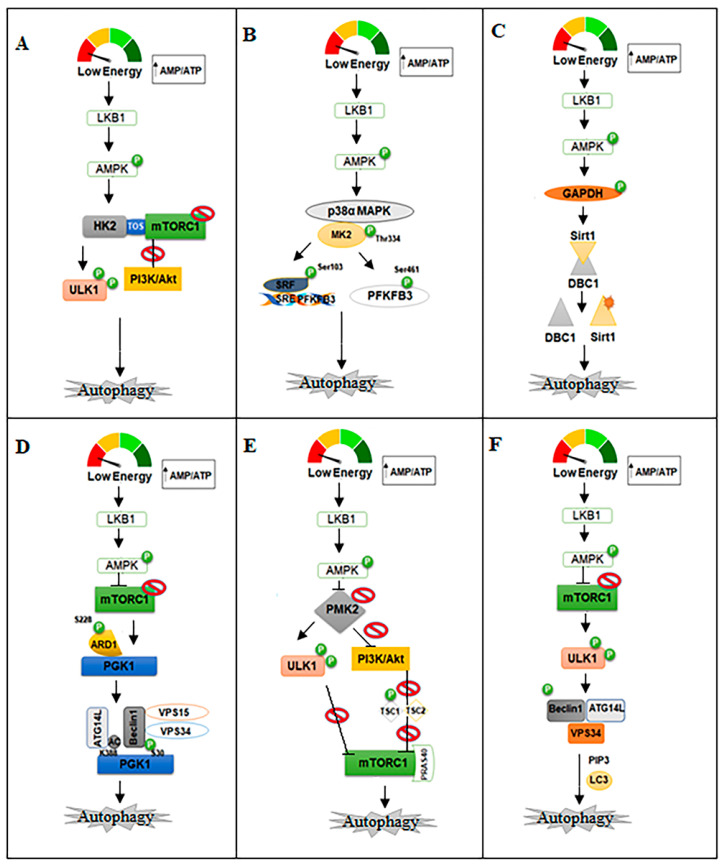
Autophagy is regulated by multiple signaling pathways, which creates a complex interaction system. The glycolysis-related enzymes and glycolysis metabolic intermediates (**A**–**F**) are involved in the regulation of autophagy, acting as a double-edged sword to sustain cancer cell survival under conditions of energy shortage.

**Figure 4 cancers-15-02195-f004:**
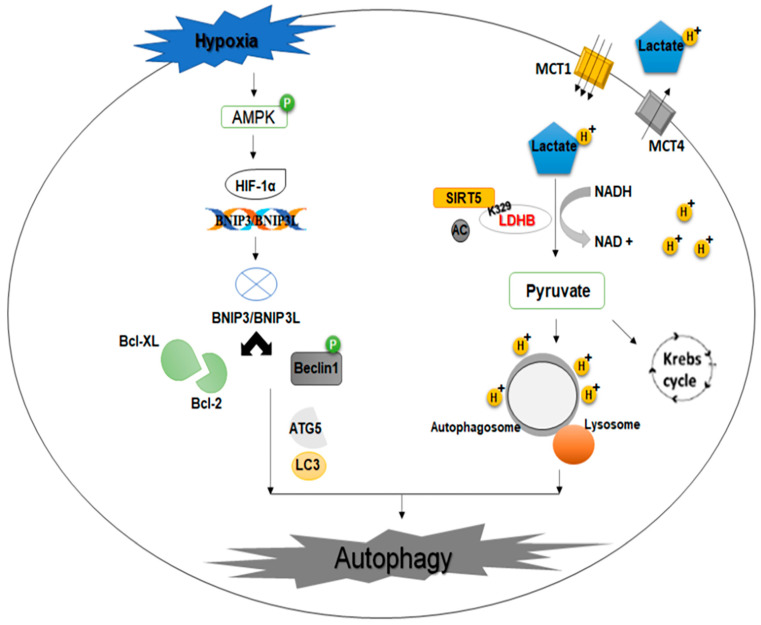
Hypoxia-induced autophagy (left): Hypoxic stress leads to the accumulation of HIF-1a. BNIP3 and BNIP3L, gene products targeted by HIF, are transcriptionally upregulated and compete with the Bcl-XL and Bcl-2 complex. This competition releases BECLIN1 from the complex, leading to the activation of autophagic machinery through the recruitment of several autophagic proteins including ATG5 and LC3. Acid-induced autophagy (right): Upon MCT1-mediated entry, lactate and NAD+ are converted into pyruvate, NADH, and H+ by LDHB. Then, the protons generated by LDHB can promote lysosomal acidification and autophagy.

**Figure 5 cancers-15-02195-f005:**
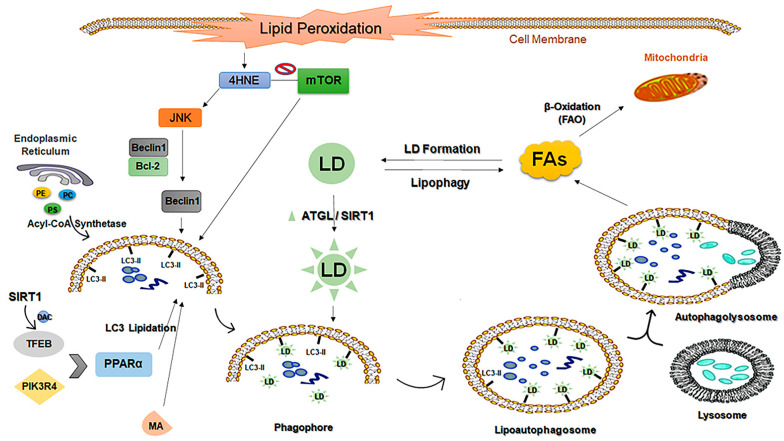
Crosstalk between lipids and autophagy. Lipids can modulate autophagy through different mechanisms.

**Figure 6 cancers-15-02195-f006:**
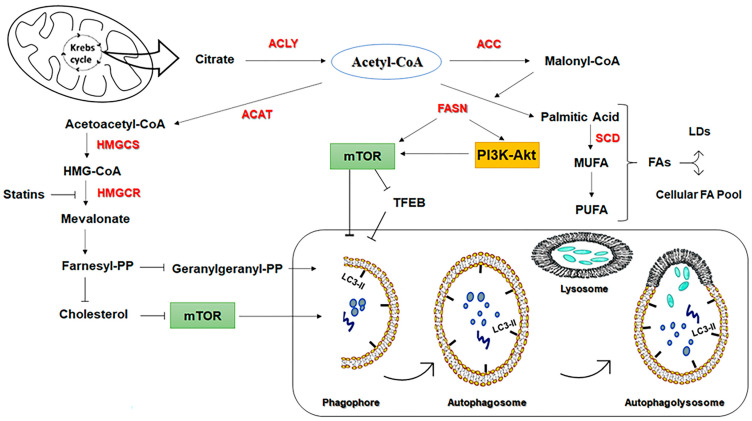
Mechanism of action of autophagy regulation upon HMGCR inhibition by statins. In addition, the mechanism by which FASN can modulate autophagy during cancer is shown. ACLY: ATP citrate lyase; ACAT: acetyl-CoA acetyltransferase; HMGCS: 3-hydroxy-3-methylglutaryl-CoA synthase; HMGCR: 3-hydroxy-3-methylglutaryl-CoA reductase; ACC: acetyl-CoA carboxylase; FASN: fatty acid synthase; SCD: stearoyl-CoA desaturase; MUFA: monounsaturated fatty acid; PUFA: polyunsaturated fatty acid.

**Figure 7 cancers-15-02195-f007:**
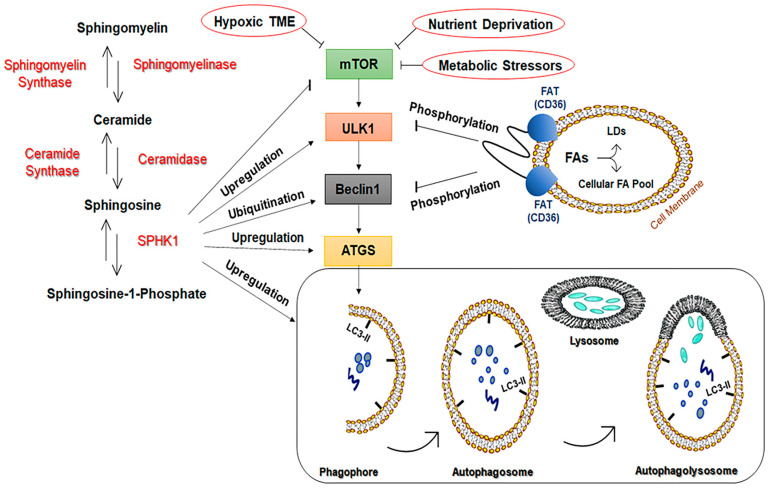
SPHK1 and FAT can regulate autophagy during cancer. A simplified scheme of sphingolipid metabolism and turnover is also shown.

**Figure 8 cancers-15-02195-f008:**
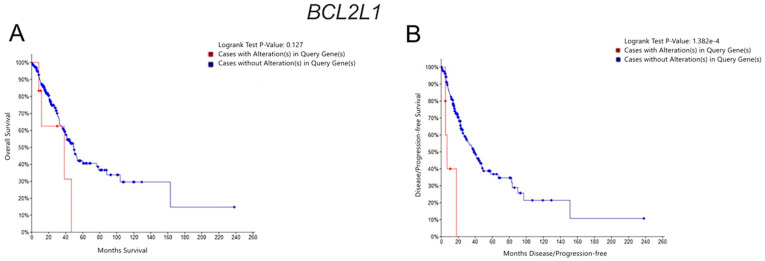
(**A**): Overall Survival Kaplan–Meier Estimate of LUAD cancer patients with BCL2L1 gene amplification (red) or without (blue). (**B**): Disease/Progression-free Kaplan–Meier Estimate in LUAD cancer patients with BCL2L1 gene amplification (red) and without (blue). Graph from Oncomine.org site using TCGA cancer data [327,328,329].

**Figure 9 cancers-15-02195-f009:**
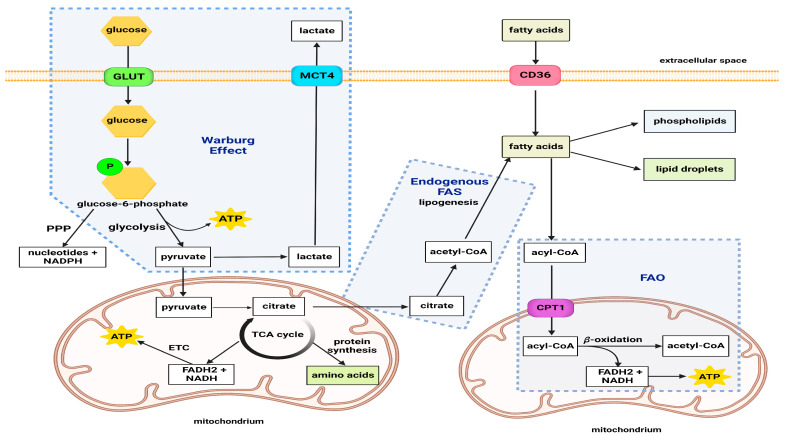
Schematic representation highlighting the metabolic pathways upregulated in cancer. In tumour cells, the uptake of glucose and aerobic glycolysis (the Warburg effect) are upregulated for rapid ATP production. The abundance of lactate that is produced is excreted into the extracellular space/tumour microenvironment. Some pyruvate is shunted into the mitochondrion for OXPHOS. The excess glucose feeds into biosynthesis, such as nucleotide, protein, and lipid synthesis, which supports uncontrolled proliferation. Altered glucose metabolism is linked to altered lipid metabolism in tumour cells, as glucose is the major substrate for lipid synthesis. Both endogenous fatty acid synthesis and fatty acid uptake are upregulated in tumour cells. The fatty acids can be subsequently used for ATP production through *β*-oxidation, which is also upregulated in tumour cells and supports uncontrolled growth and proliferation. ATP, adenosine triphosphate; CPT1, carnitine palmitoyl transferase 1; ETC, electron transport chain; FAO, fatty acid oxidation; FAS, fatty acid synthesis; GLUT, glucose transporter; MCT4, monocarboxylate transporter 4; PPP, pentose phosphate pathway; TCA cycle, tricarboxylic acid cycle.

**Table 1 cancers-15-02195-t001:** Glycolysis-related enzymes involved in the regulation of autophagy.

Enzyme Name	Enzyme Isoform	Type of Cancer	Molecules	Roles in Glycolysis and Autophagy	Ref.
**PFKFBs**	PFKFB3	Colon Adenocarcinoma	SiRNA and 3PO	Knockdown of PFKFB3 induces autophagy.	[123]
**GAPDH**	-	Pancreatic Adenocarcinoma	Genipin and Everolimus/3-MA	Induces Caspase-mediated apoptosis and inhibits BECLIN1-mediated autophagy.	[136]
**PGK**	PGK1	Breast Cancer	shRNA	Reduces the Warburg effect and reduces invasion.	[187]
**PK**	PKM2	AML	SiRNA	Reduces BECLIN1-mediated autophagy.	[150]
**GAPDH**	-	Gastric Cancer	miR-133b	Reduces glycolysis and induces autophagy by switching the PKM isoform expression from PKM2 to PKM1.	[188]
**LDH**	LDHA	Breast Cancer	SiRNA	Regulates BECLIN1-mediated autophagy.	[160]
**PK**	LDHB	Colorectal carcinoma	GW5074	Reduces autophagy by SIRT5 knockout or the inhibition of LDHB acetylation at Lys329.	[189]
**HIF**	HIF-1α	Human Liver and Glioma Cancer	shRNA- lincRNA-p21	Reduces glycolysis and inhibits autophagy by downregulating the HIF-1 protein.	[185]
**LDH**	LDHA	Cervical Cancer	SiRNA and 2-deoxyD-glucose or koningic acid	Inhibits glycolysis and reduces autophagy.	[72]
**Lactate**	-	Colorectal Carcinoma	Osimertinib	Induces autophagy by regulating the expression of MCT1 and activating LKB1/AMPK signaling.	[178]
**HIF**	HIF-1-α	Breast Cancer	7acc1	Induces autophagy via the inhibition of MCT4.	[179]
**HK**	HK2	Colon, breast, prostate	miR-143	Induces autophagy and induces proliferation.	[91,92,93,94,95,96,97]

**Table 2 cancers-15-02195-t002:** The effect of enzymes involved in lipid metabolism on different steps of autophagy.

Enzyme	Function	Effect on Autophagy		Cancer Type	Ref.
**Fatty Acid Synthase (FASN)**	Synthesizes palmitate from acetyl-CoA and malonyl-CoA	(a) Activation of mTOR and subsequent downregulation of TFEB	Inhibition	Acute Myeloid Leukemia	[196]
		(b) Activation of PI3K-Akt	Inhibition	Osteosarcoma	[198]
		(c) MAPK inactivation	Inhibition	Colorectal Cancer	[199]
**3-Hydroxy-3-MethylGlutaryl-CoA Reductase (HMGCR)**	Synthesizes cholesterol, geranylgeranyl pyrophosphate and farnesyl pyrophosphate, etc.	(a) Cholesterol-dependent inhibition of Akt-mTOR upon HMGCR inhibition by Lovastatin	Induction	Acute Lymphocytic Leukemia	[204]
		(b) Geranylgeranyl pyrophosphate-dependent upregulation of LC3-II upon HMGCR inhibition by Atorvastatin	Induction	Prostate Cancer, Rhabdomyosarcoma	[207]
**Sphingosine-1-Kinase (SPHK1)**	Synthesizes sphingosine 1 phosphate (S1P) from sphingosine	(a) Promotes biogenesis of autophagosomes by modulating the expression levels of LC3	Induction	Neuroblastoma	[223]
		(b) Increases the formation of LC3-positive autophagosomes	Induction	Breast Cancer	[224]
		(c) Activates SphK1/ERK/p ERK leading to the inactivation of mTOR, and elevated expression of ATG5 and ULK1	Induction	Colon Cancer	[225]
		(d) Enhances lysosomal degradation, increasing the ubiquitination of BECLIN1	Induction	Hepatocellular Carcinoma	[226]
		(e) Upregulation of LC3 expression	Induction	Gastric Cancer	[227]
**Fatty Acid Translocase (FAT/CD36)**	Long-chain fatty acid uptake	(a) Phosphorylation of BECLIN1 and ULK1	Inhibition	Hepatocellular Carcinoma	[229]

**Table 3 cancers-15-02195-t003:** Regulation of autophagy by lipid metabolites during cancer.

Lipid Metabolite	Effect on Autophagy		Cancer Type	Ref.
**Phosphatidic Acid**	Induction of autophagosome curvature, inhibiting mTOR	Induction	Breast Cancer	[232]
**Ceramide**	Inhibition of Akt/mTOR; Upregulation of BECLIN1 expression	Induction	Breast Cancer, Neuroblastoma	[237,239]
	Upregulation of BECLIN1 expression	Induction	Colon Cancer	[239]
	Dissociation of BCL2 from BECLIN1	Induction	Cervical Cancer	[238]
**Sphingosine 1-phosphate (S1P)**	mTOR inhibition	Induction	Prostate Cancer	[242]
	mTOR inhibition	Induction	Breast Cancer	[224,241]
**Peroxisome Proliferator-Activating Receptors (PPARs)**	PTEN activation; mTOR inhibition	Induction	Colorectal Cancer	[249]
	Increase in lysosomal acidification	Induction	Breast Cancer	[250]
**Palmitic Acid**	Activation of PKC, increasing the diacylglycerol levels	Induction	Hepatocarcinoma	[254]
**Docosahexaenoic acid (DHA)**	AMPK activation resulting in the inhibition of mTOR	Induction	p53-wild type prostate cancer	[264]
	Producing mitochondrial reactive oxygen species (ROS) resulting in Akt and mTOR inhibition	Induction	p53-mutant prostate cancer	[264]
	Upregulation of LC3 and ATG14 expression	Induction	Colon Cancer	[267]
	mTOR inhibition	Induction	Glioblastoma	[264]
	mTOR inhibition	Induction	Cervix Carcinoma	[264]
**Docosahexaenoyl ethanolamine (DHEA)**	Upregulation of PPARγ, which upregulates PTEN and, consequently, inhibits Akt-mTOR; promotes BCL2 phosphorylation and, therefore, its dissociation from BECLIN1	Induction	Breast Cancer	[270]
**Cholesterol**	mTOR inhibition	Induction	Acute Lymphocytic Leukemia, Chronic Lymphocytic Leukemia	[204]

**Table 4 cancers-15-02195-t004:** The effect of enzymes involved in glycolysis on autophagy in lung cancer.

Enzyme Name	Enzyme Isoform	Type of Cancer	Molecules	Roles in Glycolysis and Autophagy	Ref.
**HK**	HK2	NSCLC	2-DG	Induce autophagy and apoptosis by increasing LC3II and cleaved caspase3 protein levels.	[373]
		NSCLC	SiRNA	Inhibit cancer cell proliferation and tumorigenesis through decreasing glycolysis and HK2 expression at both the mRNA and protein levels.	[374]
		NSCLC	miR-143	Inhibit glycolysis and autophagy by the knockdown of HK2 and Atg2B.	[375]
**PFKFBs**	PFKFB3	NSCLC	Erlotinib drives	Reduce the EGF-mediated increase in glycolysis.	[376]
	PFKFB4	SCLC	Ibrutinib	Inhibit autophagy by targeting Erk.	[377]
**GAPDH**	-	NSCLC	SiRNA	Reduce glycolysis and cease tumor cell proliferation.	[131]
**PK**	PKM2	NSCLC	pshRNA	Induce autophagy and apoptosis.	[378]
**LDH**	LDHA	Lung and Breast Cancer	WZB117 and ShRNA	Reduce glycolysis and the induction of apoptosis.	[379]
**HIF**	HIF-1	NSCLC	SP600125	Inhibit hypoxia-induced autophagy by targeting the BECLIN1 protein.	[379]
		NSCLC	Chloramphenicol	Inhibit hypoxia-induced autophagy.	[380]
**Lactate**	-	NSCLC	SiRNA	Reduce pH-stimulated autophagy by the knockdown of the BECLIN1 protein and ER stress marker protein (GRP78).	[365]

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
