# Peer review of "Regulation of Autophagy via Carbohydrate and Lipid Metabolism in Cancer"

_cancers, 2023, doi:10.3390/cancers15082195_

Round 1
Reviewer 1 Report
This review by Alizadeh et al provides a broad insight on the crosstalk between the autophagic process and carbohydrate and lipid metabolism. Overall, the work is well structured and written and the different sections are comprehensible and exhaustive.
Some small changes may be performed to facilitate the readers in following the development of the study. My suggestion is to describe in detail, in the introduction, concepts that are frequently referred to in the text, such as the Warburg Effect and the autophagy-mediated cell death.
Table 1 would appear more complete by adding the Hexokinase 2 to the enzymes list since it is the only one missing from section 2.
I also recommend to thoroughly check the literature that is cited in the tables. For example, reference 119 in table 1 shows that knockdown of PFKB3 causes autophagy induction, as also stated in 2.2 chapter.
Moreover, some of the papers quoted in this table cannot be found in the corresponding text section (such as reference 182).
Generally, I would advise to try to improve alignment in all the tables.
Author Response
This review by Alizadeh et al provides a broad insight on the crosstalk between the autophagic process and carbohydrate and lipid metabolism. Overall, the work is well structured and written and the different sections are comprehensible and exhaustive.
Some small changes may be performed to facilitate the readers in following the development of the study. My suggestion is to describe in detail, in the introduction, concepts that are frequently referred to in the text, such as the Warburg Effect and the autophagy-mediated cell death.
Answer: The authors appreciate the wise comments of the respected reviewer which significantly improved the article and made it easier for the readers for following up. We added a short section introducing “Autophagy-Mediated Cell Death” (page 3; lines: 87-91) and “Warburg Effect” (page 5; lines: 122-141) to explain these concepts to the readers.
Table 1 would appear more complete by adding the Hexokinase 2 to the enzymes list since it is the only one missing from section 2.
Answer: The authors have added HK2 to table 1.
I also recommend to thoroughly check the literature that is cited in the tables. For example, reference 119 in table 1 shows that knockdown of PFKB3 causes autophagy induction, as also stated in 2.2 chapter.
Answer: We have checked all content and references in the table 1 and corrected them (red font).
Moreover, some of the papers quoted in this table cannot be found in the corresponding text section (such as reference 182).
Generally, I would advise to try to improve alignment in all the tables.
Answer: We have checked all content and references in the table 1 and corrected them (red font). We also adjust table 1.
Reviewer 2 Report
In this manuscript, Javad Alizadeh and colleagues give an extended overview about the changes in glycolytic and lipid biosynthetic pathways in mammalian cells and their impact on carcinogenesis via the autophagy pathway. Furthermore, they focused the impact of these metabolic pathways on autophagy in lung cancer.
Following, you will find some personal suggestion that can be used for improving the manuscript.
As reported above, authors focus on lung cancer. Why did they make this decision? If they want to only describe this disease, they should change the body of the review (and also title and abstract) and give relevance only to the contribution of glycolytic and lipid biosynthetic pathways in lung cancer. Otherwise, they should insert other cancer types in which such pathways are relevant.
At the end of the manuscript authors focus the attention of the importance of the metabolic switch between oxidative phosphorylation and glycolysis. However, throughout the manuscript the importance of mitochondria in cancer is not provided. Please insert a section regarding this.
In line with this, I consider the sections regarding lung cancer well performed. This is not for the Introduction section:
- Introduction: the part of autophagy should be improved, and authors should also insert part of the molecular pathways that regulate autophagy (such as MOR, ULK, AMPK, only to cite a few). Furthermore, authors should also mention the importance of mitophagy in regulating the glycolytic and lipid biosynthetic pathways.
At line 85, authors start to describe metabolism. I suggest to separate this from the introduction or making a sub-chapter of the introduction. Here authors also describe the Warburg effect. However, they lack to well describe the importance of mitochondrial metabolism and of OXPHOS.
Finally, the figures should be improved:
- Figure 2: this is a classic representation of glycolysis that can be found in a scholar book. Please insert the glycolysis in the context of the review and do not only give schematic representation of this pathway.
- Figure 3: This figure can be very useful for a reader. However, in this version is a little bit too confusing. I suggest to separate the A-F signaling pathway in independent panels.
Minor points:
Section 2.7. Please specify which isoform of HIF is under discussion.
Author Response
In this manuscript, Javad Alizadeh and colleagues give an extended overview about the changes in glycolytic and lipid biosynthetic pathways in mammalian cells and their impact on carcinogenesis via the autophagy pathway. Furthermore, they focused the impact of these metabolic pathways on autophagy in lung cancer.
Following, you will find some personal suggestion that can be used for improving the manuscript.
As reported above, authors focus on lung cancer. Why did they make this decision? If they want to only describe this disease, they should change the body of the review (and also title and abstract) and give relevance only to the contribution of glycolytic and lipid biosynthetic pathways in lung cancer. Otherwise, they should insert other cancer types in which such pathways are relevant.
Answer: We appreciate the important point by the respected reviewer. We have an introduction providing the reasons why we the discussion and focus on lung cancer (page 22; lines: 705-715). We have emphasized the lung cancer as it is the highest frequencies in the world and also has the highest dead rate in the world. We added a specific section to focus on lung cancer. Also, lung cancer is a major research focus for several of the authors. We also discuss lipid and glycolysis and autophagy in lung cancer in sections 4.3 and 4.4 in a detailed and specific way.
At the end of the manuscript authors focus the attention of the importance of the metabolic switch between oxidative phosphorylation and glycolysis. However, throughout the manuscript the importance of mitochondria in cancer is not provided. Please insert a section regarding this.
Answer: We added two sections (page 6; lines: 182-200) and (page 34; lines: 1151-1170) to explain the impact of mitochondria and mitophagy to cancer.
In line with this, I consider the sections regarding lung cancer well performed. This is not for the Introduction section:
- Introduction: the part of autophagy should be improved, and authors should also insert part of the molecular pathways that regulate autophagy (such as MOR, ULK, AMPK, only to cite a few). Furthermore, authors should also mention the importance of mitophagy in regulating the glycolytic and lipid biosynthetic pathways.
Answer: We added a section explaining more about the general autophagy pathway (page 2 and 3; lines: 72-76).
At line 85, authors start to describe metabolism. I suggest to separate this from the introduction or making a sub-chapter of the introduction. Here authors also describe the Warburg effect. However, they lack to well describe the importance of mitochondrial metabolism and of OXPHOS.
Answer: We added subsection (General Metabolism) to the introduction. We also added a section about mitochondria and cancer in the (General Metabolism) (page 6; line 182-200).
Finally, the figures should be improved:
- Figure 2: this is a classic representation of glycolysis that can be found in a scholar book. Please insert the glycolysis in the context of the review and do not only give schematic representation of this pathway.
Answer: The authors agree with the respected reviewers. However, we want the review paper to also be suitable for early graduate and under graduate trainees as well and we have included Figure 2 to remind everyone about the general metabolism pathways and their interconnections. In addition, we gathered all of these pathways into a single figure so it is easy for the follow up.
- Figure 3: This figure can be very useful for a reader. However, in this version is a little bit too confusing. I suggest to separate the A-F signaling pathway in independent panels.
Answer: The authors follow the respected reviewer comment and split the Figure 3 for better understanding.
Minor points:
Section 2.7. Please specify which isoform of HIF is under discussion.
Answer: It is HIF-1α and we have corrected it in section 2.7.
Round 2
Reviewer 2 Report
Dear Authors,
I appreciated you considered all my suggestions.
You greatly improved the manuscript.
Thanks.